# VARIATIONAL RECTIFIED FLOW MATCHING

## ABSTRACT

We study Variational Rectified Flow Matching, a framework that enhances classic rectified flow matching by modeling multi-modal velocity vector-fields. At inference time, classic rectified flow matching 'moves' samples from a source distribution to the target distribution by solving an ordinary differential equation via integration along a velocity vector-field. At training time, the velocity vector-field is learnt by linearly interpolating between coupled samples one drawn from the source and one drawn from the target distribution randomly. This leads to "ground-truth" velocity vector-fields that point in different directions at the same location, i.e., the velocity vector-fields are multi-modal/ambiguous. However, since training uses a standard mean-squared-error loss, the learnt velocity vector-field averages "ground-truth" directions and isn't multi-modal. Further, averaging leads to integration paths that are more curved while making it harder to fit the target distribution. In contrast, the studied variational rectified flow matching is able to capture the ambiguity in flow directions. We show on synthetic data, MNIST, and CIFAR-10 that the proposed variational rectified flow matching leads to compelling results with fewer integration steps.

## 1 INTRODUCTION

Diffusion models (Ho et al., 2020; Song et al., 2021a;b) and more generally flow matching (Liu et al., 2023; Lipman et al., 2023; Albergo & Vanden-Eijnden, 2023; Albergo et al., 2023) have been remarkably successful in recent years. These techniques have been applied across domains from computer vision (Ho et al., 2020) and robotics (Kapelyukh et al., 2023) to computational biology (Guo et al., 2024) and medical imaging (Song et al., 2022).

Flow matching (Lipman et al., 2023; Liu et al., 2023; Albergo & Vanden-Eijnden, 2023) can be viewed as a continuous time generalization of classic diffusion models (Albergo et al., 2023; Ma et al., 2024). Those in turn can be viewed as a variant of a hierarchical variational auto-encoder (Luo, 2022). At inference time, flow matching 'moves' a sample from a source distribution to the target distribution by solving an ordinary differential equation via integration along a velocity vector-field. To learn this velocity vector-field, at training time, flow matching aims to regress/match a constructed vector-field/flow connecting any sample from the source distribution — think of the data-space positioned at time zero — to any sample from the target distribution attained at time one. Notably, in a 'rectified flow,' the samples from the source and target distribution are connected via a straight line as shown in Fig. 1(a). Inevitably, this leads to ambiguity, i.e., flows pointing in different directions at the same location in the data-space-time-space domain, as illustrated for a one-dimensional data-space in Fig. 1(a). Since classic rectified flow matching employs a standard squared-norm loss to compare the predicted velocity vector-field to the constructed velocity vector-field, it does not capture this ambiguity. Hence, rectified flow matching aims to match the source and target distribution in alternative ways, leading to flow trajectories that are more complex and usually more curved. This is illustrated in Fig. 1(b).

To enable rectified flow matching to capture this ambiguity in the data-space-time-space domain, we study *variational rectified flow matching*. Intuitively, variational rectified flow matching introduces a latent variable that permits to disentangle ambiguous flow directions at each location in the data-space-time-space domain. This approach follows the classic variational inference paradigm underlying expectation maximization or variational auto-encoders. Indeed, as shown in Fig. 1(c), variational rectified flow matching permits to model flow trajectories that intersect. This results in trajectories

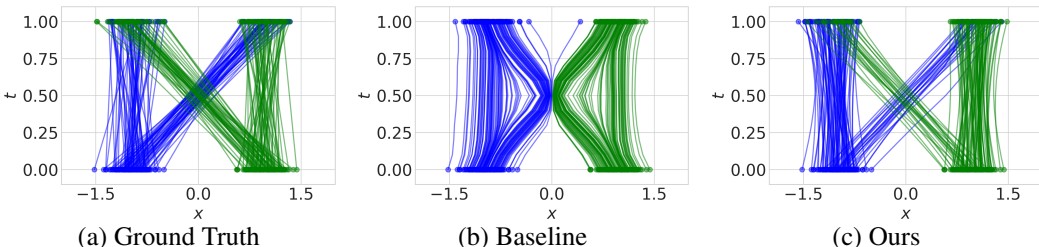

(a) Ground Truth  (b) Baseline  (c) Ours

Figure 1: Intuition and motivation: Rectified flow matching randomly couples source data and target data samples, as illustrated in panel (a). This leads to velocity vector-fields with ambiguous directions. Panel (b) shows that the classic rectified flow matching averages ambiguous targets, which leads to curved flows. In contrast, the proposed variational rectified flow matching is able to successfully model ambiguity which leads to less curved flows as depicted in panel (c).

that are more straight, which can often be integrated more quickly. Moreover, the latent variable can also be used to disentangle different directions.

Note that flow matching, diffusion models, and hierarchical variational auto-encoders are all able to capture ambiguity in the data-space, as one expects from a generative model. Importantly, variational rectified flow matching differs in that it enables to also model ambiguity in the data-space-time-space domain. This enables different flow directions at the same data-space-time-space point, allowing the resulting flows to intersect at that location.

We demonstrate the benefits of variational rectified flow matching on synthetic data, MNIST, as well as CIFAR-10. On synthetic data we show that the method leads to more accurately modeled distributions when using fewer integration steps, a property which we can empirically attribute to flow fields that are less curved. On CIFAR-10 we demonstrate that the proposed approach leads to an FID score that's on par with or slightly better than classic rectified flow matching, particularly when using fewer integration steps.

In summary, our contribution is as follows: we study the properties of variational rectified flow matching, and, along the way, offer an alternative way to interpret the flow matching procedure. We think the proposed method naturally extends classic rectified flow matching.

## 2 PRELIMINARIES

Given a dataset $\mathcal{D} = \{(x_1)\}$ comprised of data samples $x_1$, e.g., an image, generative models learn a distribution $p(x_1)$, often by maximizing the likelihood. In the following we discuss how this distribution is learnt with variational auto-encoders and rectified flow matching, and how ambiguity is captured in the data domain.

### 2.1 VARIATIONAL AUTO-ENCODERS (VAEs)

Variational inference generally and variational auto-encoders (VAEs) (Kingma & Welling, 2014) specifically have been shown to capture ambiguity. This is achieved by introducing a latent variable $z$. At inference time, a latent $z$ is obtained by sampling from the prior distribution $p(z)$, typically a zero mean unit covariance Gaussian. A decoder is then used to characterize a distribution $p(x_1|z)$ over the output space, from which an output space sample $x_1$ is obtained.

At training time, variational auto-encoders use an encoder to compute an approximate posterior distribution $q_\phi(z|x_1)$ over the latent space. As the approximate posterior distribution is only needed at training time, the data $x_1$ can be leveraged. Note, the approximate posterior distribution is often a Gaussian with parameterized mean and covariance. A sample from this approximate posterior distribution is then used as input in the distribution $p_\theta(x_1|z)$ characterized by the decoder. The loss encourages a high probability of the output space samples while pushing the approximate posterior distribution $q_\phi(z|x_1, c)$ to not deviate much from the prior distribution $p(z)$. To achieve this, formally, VAEs maximize a lower-bound on the log-likelihood, i.e.,

$$\mathbb{E}_{x_1 \sim \mathcal{D}} \log p(x_1) \geq \mathbb{E}_{x_1 \sim \mathcal{D}} \left[ \mathbb{E}_{z \sim q_\phi} \left[ \log p_\theta(x_1|z) \right] - D_{\text{KL}}(q_\phi(\cdot|x_1)|p(\cdot)) \right],$$

when training a variational auto-encoder.

## 2.2 Rectified Flow Matching

For flow matching, at inference time, a source distribution $p_0(x_0)$ is queried to obtain a sample $x_0$. This is akin to sampling of a latent variable from the prior in VAEs. Different from VAEs which perform a single forward pass through the decoder, in flow matching, the source distribution sample $x_0$ is used as the boundary condition for an ordinary differential equation (ODE). This ODE is 'solved' by pushing the sample $x_0$ forward from time zero to time one via integration along a trajectory specified via a learned velocity vector-field $v_\theta(x_t, t)$ defined at time $t$ and location $x_t$, and commonly parameterized by deep net weights $\theta$. Note, the velocity vector-field is queried many times during integration. The likelihood of a data point $x_1$ can be assessed via the instantaneous change of variables formula (Chen et al., 2018; Song et al., 2021b; Lipman et al., 2023),

$$\log p_1(x_1) = \log p_0(x_0) + \int_1^0 \operatorname{div} v_\theta(x_t, t) dt, \tag{1}$$

which is commonly (Grathwohl et al., 2018) approximated via the Skilling-Hutchinson trace estimator (Skilling, 1989; Hutchinson, 1990). Here, $\operatorname{div}$ denotes the divergence vector operator.

Intuitively, by pushing forward samples $x_0$, randomly drawn from the source distribution $p_0(x_0)$, ambiguity in the data domain is captured as one expects from a generative model.

At training time the parametric velocity vector-field $v_\theta(x_t, t)$ needs to be learnt. For this, coupled sample pairs $(x_0, x_1)$ are constructed by randomly drawing from the source and the target distribution, usually independently from each other. A coupled sample $(x_0, x_1)$ and a time $t \in [0, 1]$ is then used to compute a time-dependent location $x_t$ at time $t$ via a function $\phi(x_0, x_1, t) = x_t$. Recall, rectified flow matching uses $x_t = \phi(x_0, x_1, t) = (1 - t)x_0 + tx_1$. Interpreting $x_t$ as a location, intuitively, the "ground-truth" velocity vector-field $v(x_0, x_1, t)$ is readily available via $v(x_0, x_1, t) = \partial\phi(x_0, x_1, t)/\partial t$, and can be used as the target to learn the parametric velocity vector-field $v_\theta(x_t, t)$. Concretely, flow matching aims to learn the parametric velocity vector field $v_\theta(x_t, t)$ by matching the target via an $\ell_2$ loss, i.e., by minimizing w.r.t. the trainable parameters $\theta$ the objective

$$\mathbb{E}_{t, x_0, x_1} \left[ \|v_\theta(x_t, t) - v(x_0, x_1, t)\|_2^2 \right].$$

Consider two different couplings that lead to different "ground-truth" velocity vectors at the same data-domain-time-domain $(x_t, t)$. The parametric velocity vector-field $v_\theta(x_t, t)$ is then asked to match/regress to a different target given the same input $(x_t, t)$. This leads to averaging and the optimal functional velocity vector-field $v^*(x_t, t) = \mathbb{E}_{\{(x_0, x_1, t): \phi(x_0, x_1, t) = x_t\}} [v(x_0, x_1, t)]$. Hence, ambiguity in the data-domain-time-domain is not captured. In the following we discuss and study a method that is able to model this ambiguity.

# 3 Variational Rectified Flow Matching

Our goal is to capture the ambiguity inherent in "ground-truth" velocity vector-fields obtained from typically used couplings $(x_0, x_1)$ that connect source distribution samples $x_0 \sim p_0$ with target data samples $x_1 \in \mathcal{D}$. Here, $p_0$ is a known source distribution and $\mathcal{D}$ is a considered dataset. This differs from classic rectified flow matching which does not capture this ambiguity even for simple distributions as shown in Fig. 1 and as discussed in Sec. 2. The struggle to capture ambiguity leads to velocity vector fields that are more curved and consequently more difficult to integrate at inference time. In turn, this leads to distributions that may not fit the data as well. We will show evidence for both, more difficult integration and less accurately captured data distributions in Sec. 4.

To achieve our goal we combine rectified flow matching and variational auto-encoders. In the following we first discuss the objective before detailing training and inference.

## 3.1 Objective

The goal of flow matching is to learn a velocity vector-field $v_\theta(x_t, t)$ that transports samples from a known source distribution $p_0$ at time $t = 0$ to samples from a commonly unknown probability density function $p_1(x_1)$ at time $t = 1$. The probability densities $p_0, p_1$ and the velocity vector-field $v_\theta$ are related to each other via the transport problem

$$\frac{\partial \log p_t(x_t)}{\partial t} = -\operatorname{div} v_\theta(x_t, t), \tag{2}$$

or its integral form given in Eq. (1).

Solving the partial differential equation given in Eq. (2) in general analytically is challenging, even when assuming availability of the probability density functions, i.e., when addressing a classic boundary value problem.

However, if we assume the probability density functions to be Gaussians and if we restrict the velocity vector-field to be constant, i.e., of the simple parametric form $v_\theta(x_t, t) = \theta$, we can obtain an analytic solution. This is expressed in the following claim:

**Claim 1** *Consider two Gaussian probability density functions $\tilde{p}_0 = \mathcal{N}(\xi_0; x_0, I)$ and $\tilde{p}_1 = \mathcal{N}(\xi_1; x_1, I)$ with mean $x_0$ and $x_1$ respectively. Let's restrict ourselves to a constant velocity vector-field $v_\theta(\xi_t, t) = \theta$. Then $\theta = x_1 - x_0$ solves the partial differential equation given in Eq. (2) and its integral form given in Eq. (1) and $x_t = (1 - t)x_0 + tx_1$.*

**Proof:** Given the constant velocity vector-field $v_\theta(\xi_t, t) = \theta$, we have $\int_1^0 \operatorname{div} v_\theta(\xi_t, t)dt \equiv 0$. Plugging this and both probability density functions into Eq. (1) yields $(\xi_0 - x_0)^2 - (\xi_1 - x_1)^2 \equiv 0$ $\forall \xi_0, \xi_1$. Using $\xi_1 = \xi_0 + \int_0^1 v_\theta(\xi_t, t)dt = \xi_0 + \theta$ leads to $(\xi_0 - x_0)^2 - (\xi_0 - x_1 + \theta)^2 \equiv 0$ $\forall \xi_0$ which is equivalent to $(x_1 - x_0 - \theta)(2\xi_0 - x_0 - x_1 + \theta) \equiv 0$ $\forall \xi_0$. This can only be satisfied $\forall \xi_0$ if $\theta = x_1 - x_0$, leading to $x_t = x_0 + t\theta = (1 - t)x_0 + tx_1$, which proves the claim. ∎

The arguably very simple setup in Claim 1 provides intuition for the objective of classic rectified flow matching and offers an alternative way to interpret the flow matching procedure. Specifically, instead of two Gaussian probability density functions $\tilde{p}_0$ and $\tilde{p}_1$, we assume the real probability density functions for the source and target data are composed of Gaussians centered at given data points $x_0$ and $x_1$ respectively, e.g., $p_0(\xi_0) = \sum_{x_0 \in \mathcal{S}} \mathcal{N}(\xi_0; x_0, I)/|\mathcal{S}|$. Moreover, importantly, let us assume that the velocity vector-field $v_\theta(x_t, t)$ at a data-domain-time-domain location $(x_t, t)$ is characterized by a uni-modal standard Gaussian

$$p(v|x_t, t) = \mathcal{N}(v; v_\theta(x_t, t), I)$$

with a parametric mean $v_\theta(x_t, t)$. Maximizing the log-likelihood of the empirical "velocity data" is equivalent to the following objective

$$\mathbb{E}_{t, x_0, x_1}\left[\log p(x_1 - x_0|x_t, t)\right] \propto -\mathbb{E}_{t, x_0, x_1}\left[\|v_\theta(x_t, t) - x_1 + x_0\|_2^2\right]. \tag{3}$$

Note that this objective is identical to classic rectified flow matching. Moreover, note our use of the standard rectified flow velocity vector-field, also derived in Claim 1.

This derivation highlights a key assumption: because the vector field is parameterized via a Gaussian at each data-domain-time-domain location, ambiguity cannot be captured: the Gaussian distribution is uni-modal. Therefore, classic rectified flow matching aims to average the "ground-truth" velocities.

As mentioned before, this can be sub-optimal. To capture ambiguity, we study the use of a mixture model over velocities at each data-domain-time-domain location. Concretely, we assume an *unobserved* continuous random variable $z$, drawn from a prior distribution $p(z)$, governs the mean of the *conditional* distribution of the velocity vector-field, i.e.,

$$p(v|x_t, t, z) = \mathcal{N}(v; v_\theta(x_t, t, z), I).$$

Note that this model captures ambiguity because $p(v|x_t, t) = \int p(v|x_t, t, z)p(z)dz$ mixes Gaussians with different means.

We now derive the variational flow matching objective. Since the random variable $z$ is not observed, at training time, we introduce a recognition model $q_\phi(z|x_0, x_1, x_t, t)$ a.k.a. an encoder. It is parameterized by $\phi$ and approximates the intractable true posterior.

Using this setup, the marginal likelihood of an individual data point can be lower-bounded by

$$\log p(v|x_t, t) \geq \mathbb{E}_{z \sim q_\phi}\left[\log p(v|x_t, t, z)\right] - D_{\text{KL}}(q_\phi(\cdot|x_0, x_1, x_t, t)|p(\cdot)). \tag{4}$$

Replacing the log-probability of the Gaussian in the derivation of Eq. (3) with the lower bound given in Eq. (4) immediately leads to the variational rectified flow matching objective $\mathbb{E}_{t, x_0, x_1}\left[\log p(x_1 - x_0|x_t, t)\right] \geq$

$$\mathbb{E}_{t, x_0, x_1}\left[-\mathbb{E}_{z \sim q_\phi}\left[\|v_\theta(x_t, t, z) - x_1 + x_0\|_2^2\right] - D_{\text{KL}}(q_\phi(\cdot|x_0, x_1, x_t, t)|p(\cdot))\right]. \tag{5}$$

---

**Algorithm 1:** Variational Rectified Flow Matching Training

---

**Data:** source distribution $p_0$ and target sample dataset $\mathcal{D}$

1 **while** *stopping conditions not satisfied* **do**

2     sample $x_0 \sim p_0, x_1 \in \mathcal{D}$;                 //we use a mini-batch

3     sample $t \sim U(0,1)$;        //different $t$ for each mini-batch sample

4     $x_t = (1 - t)x_0 + tx_1$;

5     get latent $z = \mu_\phi(x_0, x_1, x_t, t) + \epsilon\sigma_\phi(x_0, x_1, x_t, t)$ with $\epsilon \sim \mathcal{N}(0,1)$;

       //reparameterization trick

6     compute loss following Eq. (5);

7     perform gradient update on $\theta$, $\phi$;

8 **end**

---

**Algorithm 2:** Variational Rectified Flow Matching Inference

---

**Data:** source distribution $p_0$

1 sample $x_0 \sim p_0$;

2 get latent $z \sim p(z)$;

3 ODE integrate $x_0$ from $t = 0$ to $t = 1$ using velocity vector-field $v_\theta(x_t, t, z)$;

---

We note that this objective could be extended in a number of ways: for instance, the prior $p(z)$ could be a trainable deep net conditioned on $x_0$ and/or $t$. Note however that this leads to a more complex optimization problem with a moving target. We leave a study of extensions to future work.

In the following we first discuss optimization of this objective before detailing the inference procedure.

### 3.2 TRAINING

To optimize the objective given in Eq. (5), we follow the classic VAE setup. Specifically, we let the prior $p(z) = \mathcal{N}(z; 0, I)$ and the approximate posterior $q_\phi(z|x_0, x_1, x_t, t) = \mathcal{N}(z; \mu_\phi(x_0, x_1, x_t, t), \sigma_\phi(x_0, x_1, x_t, t))$. This enables analytic computation of the KL-divergence in Eq. (5). Note that the mean of the approximate posterior is obtained from the deep net $\mu_\phi(x_0, x_1, x_t, t)$ and the standard deviation is obtained from $\sigma_\phi(x_0, x_1, x_t, t)$. Further, we use the re-parameterization trick to enable optimization of the objective w.r.t. the trainable parameters $\theta$ and $\phi$. Moreover, we use a single-sample estimate for the expectation over the unobserved variable $z$. We summarize the training procedure in Algorithm 1. Note, it's more effective to work with a mini-batch of samples rather than a single data point, which was merely used for readability in Algorithm 1.

Note that variational rectified flow matching training differs from training of classic rectified flow matching in only a single step: computation of a latent sample $z$ in Line 5 of Algorithm 1. From a computational point of view we add a deep net forward pass to obtain the mean $\mu_\phi$ and standard deviation $\sigma_\phi$ of the approximate posterior, and a backward pass to obtain the gradient w.r.t. $\phi$. Also note that the velocity vector-field architecture $v_\theta(x_t, t, z)$ might be more complex as the latent variable $z$ needs to be considered. The additional amount of computation is likely not prohibitive.

We provide implementation details for the deep nets $v_\theta(x_t, t, z)$, $\mu_\phi(x_0, x_1, x_t, t)$, and $\sigma_\phi(x_0, x_1, x_t, t)$ in Sec. 4, as their architecture depends on the data.

### 3.3 INFERENCE

We summarize the inference procedure in Algorithm 2. Note that we draw a latent variable only once prior to classic ODE integration of a random sample $x_0 \sim p_0$ drawn from the source distribution $p_0$. To obtain the latent $z$ we sample from the prior $z \sim p(z) = \mathcal{N}(z; 0, I)$. Subsequently, we ODE integrate the velocity field $v_\theta(x_t, t, z)$ from time $t = 0$ to time $t = 1$ starting from a random sample $x_0$ drawn from the source distribution.

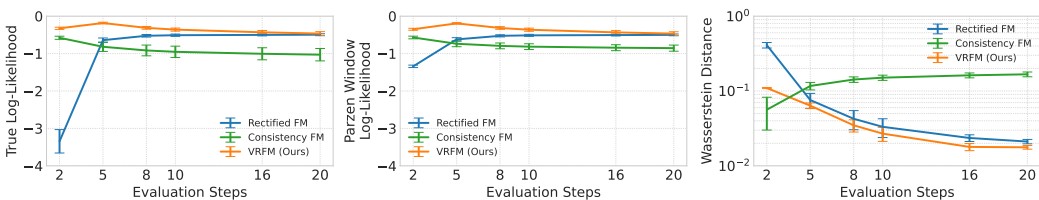

Figure 2: Quantitative evaluation on synthetic 1D data for varying evaluation steps. Metrics are averaged over three runs with different random seeds. For True and Parzen Window Log-Likelihood, higher values are better. For Wasserstein Distance, lower values are preferred.

# 4 EXPERIMENTS

We evaluate the efficacy of variational rectified flow matching and compare to the classic rectified flow (Lipman et al., 2023; Liu et al., 2023; Albergo & Vanden-Eijnden, 2023) and the recent consistency flow matching (Yang et al., 2024) using multiple datasets. We show the benefits of our method in modeling the velocity ambiguity in the data-domain-time-domain, which leads to compelling results with fewer integration steps using ODE solvers. Moreover, we demonstrate that explicitly modeling ambiguity through a conditional latent $z$ enhances the interpretability of flow matching models, leading to controlability.

## 4.1 SYNTHETIC 1D DATA

For our synthetic 1D experiments, the source distribution is a zero-mean, unit-variance Gaussian, while the target distribution is bimodal, with modes centered at $-1.0$ and $1.0$.

For the rectified flow matching and the consistency flow matching baselines, we use a multi-layer MLP network $v_\theta$ to model the velocity. The network operates on inputs $x_t$ and $t$ and predicts the velocity through a series of MLP layers. We follow this structure in our variational rectified flow matching, but add an encoding layer for the latent variable $z$. The posterior model $q_\phi$ follows a similar design as $v_\theta$, outputting $\mu_\phi$ and $\sigma_\phi$. At inference time, $q_\phi$ isn't used. Instead, we sample directly from the prior distribution $p(z) = \mathcal{N}(z; 0, I)$. We set the the KL loss weight to $1.0$. Further implementation details are provided in Appendix A.

We assess the performance using the Euler ODE solver and vary the evaluation steps. Results are presented in Fig. 2. Across all metrics, i.e., True Log-Likelihood, Parzen Window Log-Likelihood, and Wasserstein Distance, and most evaluation steps, our method outperforms both baselines. Notably, as the model handles ambiguity in the data-domain-time-domain, it produces reasonable results even for 2 or 5 evaluation steps. Qualitative visualizations of flow trajectories are provided in Appendix B.

To better understand the velocity ambiguity and to assess the efficacy of our model in handling it, we randomly sample different trajectories and plot the velocity range standard deviation across predefined bins in the data-domain-time-domain, as shown in Fig. 3. The ground-truth flow in Fig. 3 (a) shows that the standard deviation increases with time, peaking at $(x = 0.0, t = 0.75)$. The velocity distribution transitions from a bi-modal distribution at early times $t$ to a uni-modal distribution at later times $t$. Fig. 3 (b) shows that the rectified flow baseline, which uses an MSE loss, fails to model the velocity distribution faithfully, collapsing to a Dirac-delta distribution as expected. This is also observed for the consistency flow matching baseline, as results in Appendix K show. In contrast, Fig. 3 (c) demonstrates that our model successfully captures the distribution with higher velocity standard deviation range, matching the ground-truth flow reasonably, albeit not perfectly.

As discussed in Section 3.2, the posterior $q_\phi$ can be conditioned in different ways. To understand the implications, we performed ablation studies and visualized the velocity distribution maps in Fig. 3 (c)-(f). For $x_0$ conditioning (d), the model struggles to predict the bi-modal distribution at early timesteps ($x_t = 0.0, t = 0.0$) due to the absence of $x_1$ information. However, when $t$ is sufficiently large, the model can infer $x_1$ from $x_t$, enabling it to predict a bimodal distribution again at $(x = 0.0, t = 0.5)$. Conversely, with $x_1$ conditioning (e), the model fails to capture the ground-truth

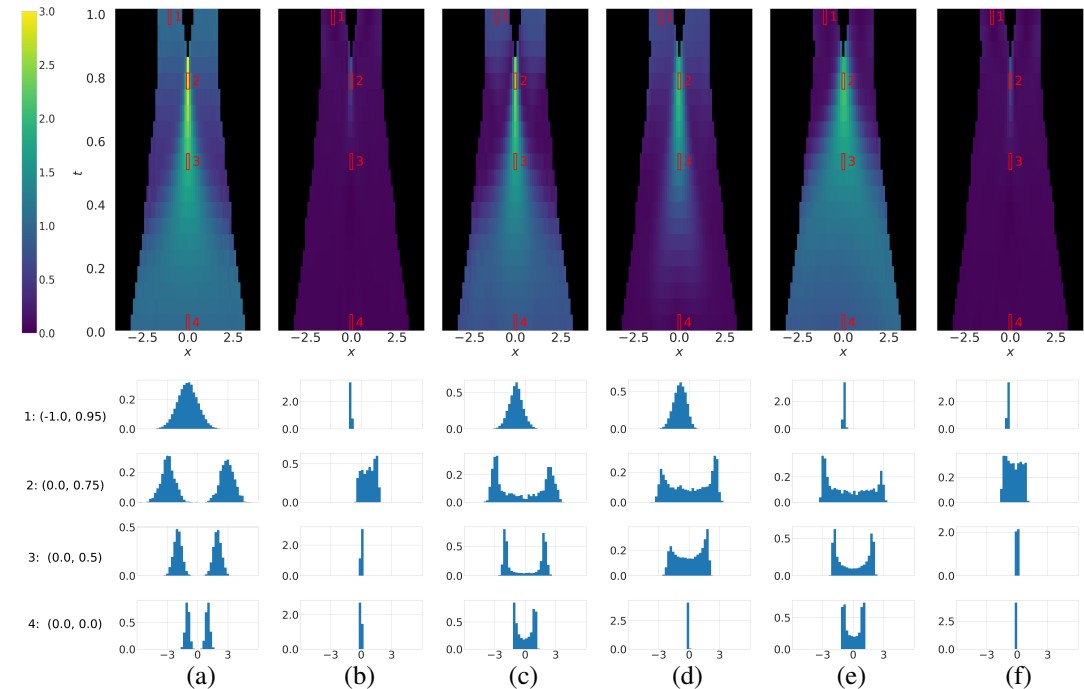

Figure 3: 1D velocity ambiguity analysis with various conditioning options and sampling strategies. (a) Ground Truth (GT), (b) Baseline (Rectified Flow), (c) Ours $(x_0 + x_1 + x_t)$, (d) Ours $(x_0)$, (e) Ours $(x_1)$, (f) Ours $(x_t)$. The heatmap illustrates the velocity standard deviation for sampled bins in data-domain-time-domain, along with histograms of the velocity at four sampled locations. Our method effectively models velocity ambiguity, while the baseline produces deterministic outputs.

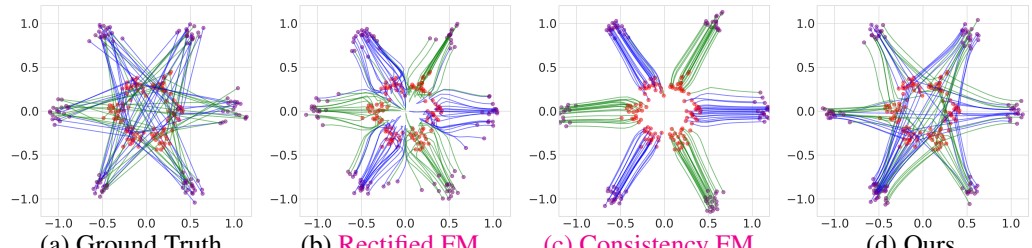

Figure 4: Flow visualization for synthetic 2D data using the Euler solver with 20 function evaluations. Sampled points from the source distribution are shown in red, and points from the target distribution in purple. Different from Rectified FM, which predicts flow trajectories with sharp curvature and U-turns to avoid crossings, and Consistency FM, which models straight lines, our model captures velocity ambiguity and predicts flows that intersect.

distribution at later timesteps $(x = -1.0, t = 0.95)$ as the influence of $x_1$ diminishes. With $x_t$ conditioning (f), the ambiguity plot follows the baseline as no extra data is provided to the posterior.

## 4.2 SYNTHETIC 2D DATA

We further test efficacy using synthetic 2D data. Following Liu et al. (2023), we model the source distribution as a mixture of Gaussian components positioned at six equidistant points along a circle with a radius of 1/3, shown in red in Fig. 4 (a). The target distribution follows a similar structure, but with components arranged along a larger circle with a radius of 1, shown in purple.

For the architecture we follow Section 4.1 and condition the posterior on $[x_0, x_1, x_t]$. We report the Parzen window log-likelihood and the true log-likelihood for various evaluation steps of the Euler ODE solver, as shown in Fig. 5. Compared to the rectified flow and consistency flow baselines, our model shows a more significant performance boost. We hypothesize that this is due to the increased complexity of the task: explicitly modeling ambiguity avoids the need for curved trajectories, making

Figure 5: Quantitative evaluation on synthetic 2D data for varying evaluation steps. Metrics are averaged over three runs with different random seeds.

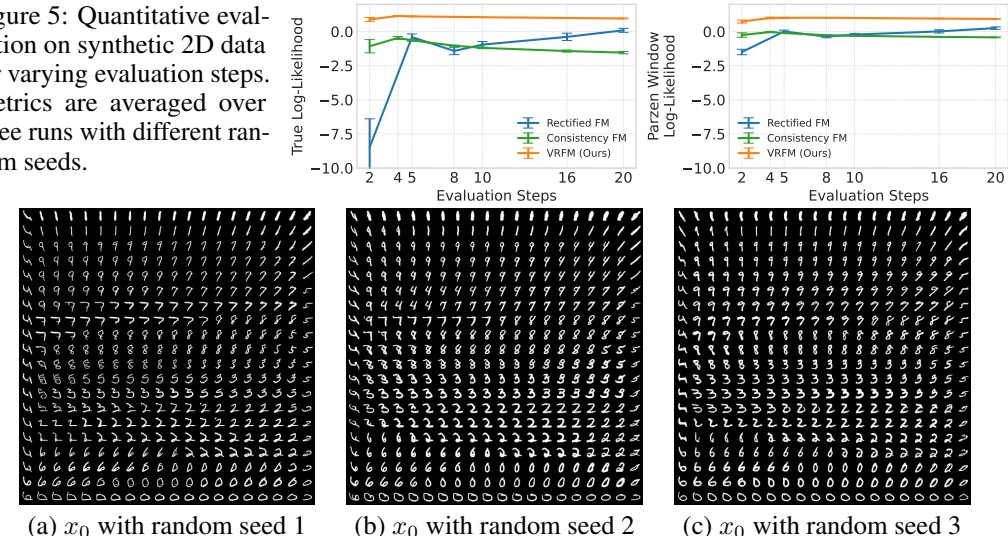

(a) $x_0$ with random seed 1  (b) $x_0$ with random seed 2  (c) $x_0$ with random seed 3

Figure 6: Visualization of learned MNIST manifold with different random noise $x_0$.

it easier to fit the target distribution. The qualitative flow visualization in Fig. 4 support this hypothesis: the rectified flow requires a U-turn to avoid collisions, while our model, aided by the variational training objective, moves in trajectories that intersect and aren't as curved.

## 4.3 MNIST

Modeling ambiguity not only improves results with fewer evaluation steps but also enables more explicit control without additional conditioning signals. We implemented a vanilla convolutional network with residual blocks (He et al., 2015) and applied variational rectified flow matching to the MNIST dataset (LeCun et al., 1998). We use $(x_0, x_1, x_t)$ as input to $q_\phi$ and set the KL loss weight to $1e^{-3}$. The detailed architecture and training paradigm is provided in Appendix C.

Following Kingma & Welling (2014), we set the latent variable $z$ to be 2-dimensional. During inference, we sample linearly spaced coordinates on the unit square, transforming them through the inverse CDF of the Gaussian to generate latents $z$. Using these latents, we integrate the samples with an ODE solver and plot the generated samples in Fig. 6. To show the effects of the source distribution sample $x_0$ and the latent $z$, we visualize the learned MNIST manifold for three randomly sampled $x_0$ values in Fig. 6 (a)-(c). The results demonstrate that the latent space $z$ enables smooth interpolation between different digits within the 2D manifold, providing control over the generated images. By adjusting $z$, we can transition between various shapes and styles. The initial noise $x_0$ enhances the generation process by introducing additional variations in character styles, allowing the model to better capture the target data distribution. We also evaluate the FID scores of our method using this 2-dimensional conditional latent space and report the results in Fig. 7. Despite the small latent dimension, it still enables the velocity model $v_\theta$ to achieve better FID scores than the baselines, except at 2 evaluation steps where Consistency FM (Yang et al., 2024) performs best.

## 4.4 CIFAR-10

Next, we evaluate our method on the CIFAR-10 data, a widely used benchmark in prior work (Lipman et al., 2023; Tong et al., 2024). For a fair comparison, we use the architecture and training paradigm of (Tong et al., 2024), but train the UNet model with the variational rectified flow loss detailed in Eq. (5). The UNet consists of downsampling and upsampling residual blocks with skip connections, and a self-attention block added after the residual block at $16 \times 16$ resolution and in the middle bottleneck layer. The model takes both $x_t$ and $t$ as input, with the time embedding $t$ used to regress learnable scale and shift parameters $\gamma$ and $\beta$ for adaptive group norm layers.

The posterior model $q_\phi$ shares similar encoder structure as $v_\theta$: image space inputs are chosen from $[x_0, x_1, x_t]$ and concatenated along the channel dimension, while time $t$ is conditioned using adaptive group normalization. The network predicts $\mu_\phi$ and $\sigma_\phi$ with dimensions $1 \times 1 \times 768$. During training,

Figure 7: FID score evaluation for the MNIST experiment conducted across three trials with different random seeds. Our model with a latent dimension of 2 outperforms the baselines, except at 2 evaluation steps where Consistency FM performs best. Note, the latent dimension of 2 is chosen for a controllability analysis rather than being optimized for FID score improvement.

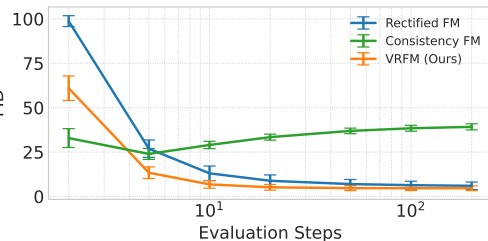

| NFE / sample | 2 | 5 | 10 | 50 | 100 | 1000 | Adaptive |
|---|---|---|---|---|---|---|---|
| OT-FM (Lipman et al., 2023; Tong et al., 2024) | 166.655 | 36.188 | 14.396 | 5.557 | 4.640 | 3.822 | 3.655 |
| I-CFM (Liu et al., 2023; Tong et al., 2024) | 168.654 | 35.489 | 13.788 | 5.288 | 4.461 | 3.643 | 3.659 |
| 1   VRFM (adaptive norm, $x_1$, 2e-3) | 135.275 | 28.912 | **13.226** | 5.382 | 4.430 | 3.642 | 3.545 |
| 2   VRFM (adaptive norm, $x_1$, 5e-3) | 159.940 | 35.293 | 14.061 | **5.265** | **4.349** | **3.582** | 3.561 |
| 3   VRFM (adaptive norm, $x_1 + t$, 5e-3) | 117.666 | 27.464 | 13.632 | 5.512 | 4.484 | 3.614 | **3.478** |
| 4   VRFM (bottleneck sum, $x_1 + t$, 2e-3) | **104.634** | **25.841** | 13.508 | 5.618 | 4.540 | 3.596 | 3.520 |

Table 1: Following Tong et al. (2024), we train the same UNet model and reported the FID scores for our method and the baselines using both fixed-step Euler and adaptive-step Dopri5 ODE solvers. The baselines checkpoint was directly taken from Tong et al. (2024). We present four model variants of our VRFM, which differ in fusion mechanism, posterior model input, and KL loss weight.

the conditional latent $z$ is sampled from the predicted posterior, and at test time, from a standard Gaussian prior. The latent is processed through two MLP layers and serves as a conditional signal for the velocity network $v_\theta$. We identify two effective approaches as conditioning mechanisms: adaptive normalization, where $z$ is added to the time embedding before computing shift and offset parameters, and bottleneck sum, which fuses the latent with intermediate activations at the lowest resolution using a weighted sum before upsampling. The detailed implementation is provided in Appendix D.

We evaluate results using FID scores computed for varying numbers of function evaluations, as shown in Table 1. Four model variants were tested, differing in fusion mechanisms, posterior model $q_\phi$ inputs, and KL loss weighting. Compared to prior work (Lipman et al., 2023; Liu et al., 2023; Tong et al., 2024), model 1 achieves superior FID scores with fewer function evaluations and performs comparably at higher evaluations. Using the adaptive Dopri5 solver further improves scores, highlighting the importance of capturing flow ambiguity. Model 2 increases the KL loss weight, improving performance at higher function evaluations but reducing effectiveness at lower evaluations, likely due to reduced information from latent $z$. Model 3, with additional time conditioning, significantly improves FID at low evaluations and performs best with the adaptive solver. Model 4, incorporating bottleneck sum fusion, delivers robust FID scores across evaluation settings, demonstrating the flexibility of the variational rectified flow objective with different fusion strategies.

Compared to other input configurations, we find conditioning on $x_1$ with optional $t$ generally yields better results. We hypothesize that this is due to the convolutional net's ability to effectively handle noisy data like $x_0$. Furthermore, the large parameter count (i.e., 38M) of the velocity network $v_\theta$ may not find any additional useful information in the latent. We leave a search for more scalable variational rectified flow conditioning mechanisms for larger models as future work.

Similar to the results reported for MNIST data in Section 4.3, we observe clear patterns in color and content for the generated samples $x_1$, demonstrating a degree of controllability. Fig. 8 visualizes three sets of images (a)–(c). Each set is conditioned on a different latent $z$, while the starting noise $x_0$ varies across individual images within each set. The same noise $x_0$ is applied to images at the same grid location across all subplots. Images conditioned on the same latent exhibit consistent color patterns, while images at the same grid location display similar content, as highlighted in the last row.

## 5 RELATED WORK

Generative modeling has advanced significantly in the last decade, thanks in part due to seminal works like generative adversarial nets (Goodfellow et al., 2014), variational auto-encoders (Kingma & Welling, 2014), and normalizing flows (Rezende & Mohamed, 2015).

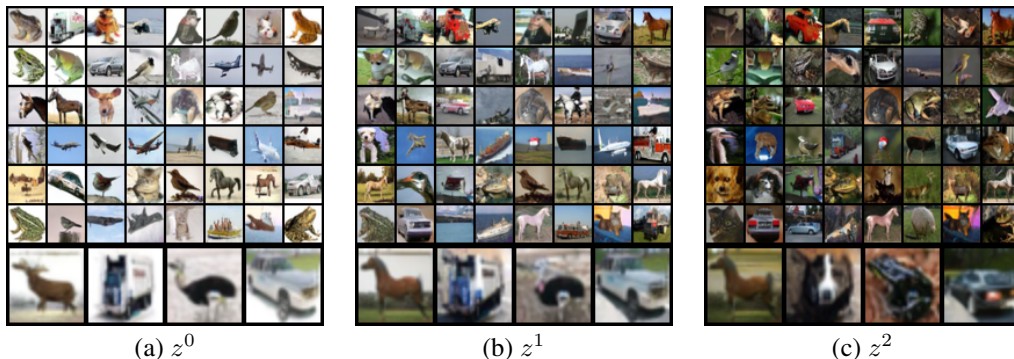

(a) $z^0$        (b) $z^1$        (c) $z^2$

Figure 8: Varying $x_0$ for a fixed latent $z$. Images at the same position across panels share the same $x_0$, while images within a panel share the same latent sampled from the prior distribution.

More recently, score matching (Song & Ermon, 2019) and diffusion models (Ho et al., 2020) were introduced. They can be viewed as augmenting variational auto-encoders hierarchically (Luo, 2022) while restricting involved distributions to be Gaussian. Notably, and analogously to classic discrete normalizing flows, the number of hierarchy levels, i.e., the number of time steps, remained discrete, which introduced complications.

Flow matching (Lipman et al., 2023) was introduced recently as a compelling alternative to avoid some of these complications. It formulates an ordinary differential equation (ODE) in continuous time. This ODE connects a source distribution to a target distribution. Solving the ODE via forward integration through time permits to obtain samples from the target distribution, essentially by 'moving' samples from the known source distribution to the target time along a learned velocity field.

To learn the velocity field, various mechanisms to interpolate between the source distribution and the target distribution have been considered (Lipman et al., 2023; Liu et al., 2023; Tong et al., 2024). Rectified flow matching emerged as a compelling variant, which linearly interpolates between samples from the two distributions. For instance, it was used to attain impressive results on large scale datasets (Ma et al., 2024; Esser et al., 2024). Compared to other interpolation techniques, linear interpolation encourages somewhat straight flows, which simplifies numerical solving of the ODE.

The importance of straight flows was further studied in ReFlow (Liu et al., 2023). Multiple ODEs are formulated and multiple velocity fields are learned one after the other by sequentially adjusting the interpolations and 're-training.' Consistency models (Song et al., 2023; Kim et al., 2023; Yang et al., 2024) strive for straight flows by modifying the loss to encourage self-consistency across timesteps. More details are provided in Appendix K.

While the aforementioned works aim to establish straight flows either via 're-training' or 're-parameterizing' of an already existing flow, differently, in this work we study the results of enabling a rectified flow to capture the ambiguity inherent in the usually employed ground-truth flow fields.

Structurally similar to this idea is work by Preechakul et al. (2022). In a first stage, an autoencoder is trained to compress images into a latent space. The resulting latents then serve as a conditioning signal for diffusion model training in a second stage. Note, this two-stage approach doesn't directly model ambiguity in the data-space-time-space domain. In similar spirit is work by Pandey et al. (2022). A VAE and a diffusion model are trained in two separate stages, with the goal to enable controllability of diffusion models. Related is also work by Eijkelboom et al. (2024) which focuses on flow matching only for categorical data, achieving compelling results on graph generation tasks.

## 6  CONCLUSION

We study Variational Rectified Flow Matching, a framework which enables to model the multimodal velocity vector fields induced by the ground-truth linear interpolation between source and target distribution samples. Encouraging results can be achieved on low-dimensional synthetic and high-dimensional image data.

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

## APPENDIX: VARIATIONAL RECTIFIED FLOW MATCHING

## A  IMPLEMENTATION DETAILS OF SYNTHETIC EXPERIMENTS

In the rectified flow baseline, the velocity network $v_\theta$ features separate encoders for time $t$ and data $x$. Each encoder consists of a sinusoidal positional encoding layer followed by two MLP layers with GeLU activation. The resulting time and data embeddings are concatenated and passed into a four-layer MLP, also utilizing GeLU activations. Both the positional embedding and hidden dimensions of the encoder and decoder are set to 64. The training batch size is 1000, and we employ the standard rectified flow objective to compute the current data $x_t = (1-t)x_0 + tx_1$ and the ground truth velocity $u_t = x_1 - x_0$, using L2 loss for supervision.

In the consistency flow matching baseline, we adopt the same velocity network $v_\theta$ and modify the loss function to incorporate the velocity consistency loss proposed by Yang et al. (2024). We find the hyperparameter settings suggested by the publicly available codebase to work best. Specifically, we use $\Delta t = 1 \times 10^{-3}$, $N_{segments} = 2$, and *boundary* $= 0.0$ for the first training stage, transitioning to *boundary* $= 0.9$ in the second stage. Additionally, the loss weighting factor $\alpha$ is set to $1 \times 10^{-5}$. For complete implementation details, we kindly direct readers to the open-source repository which we used to obtain the reported results.[1]

For both baselines, the AdamW optimizer is used with the default weight decay and a learning rate of $1 \times 10^{-3}$, over a total of 20,000 training iterations.

In our variational flow matching approach, the velocity network $v_\theta$ incorporates an additional latent encoding module comprising three MLP layers with a hidden dimension of 128. The conditional latent embedding $z$ is concatenated with the embeddings for time $t$ and data $x$. The decoder maintains the same structure as the baseline, with the first MLP layer adjusted to accommodate the increased channel input. For the posterior model $q_\phi$, we employ a similar architecture, designing a separate encoder for each possible input selected from $[x_0, x_1, x_t, t]$. Each encoder consists of a sinusoidal positional encoder layer followed by two MLP layers with GeLU activation. The output embeddings are concatenated along the channel dimension and processed through three MLP layers to produce the predicted $\mu_\phi$ and $\sigma_\phi$. The latent dimension of $z$ is set to 4 for 1D experiment and 8 for 2D experiment. During training, we utilize the reparameterization trick to sample $z$ from the predicted posterior distribution; during inference, the posterior model $q_\phi$ is omitted, and sampling is performed from a unit variance Gaussian prior distribution. The loss is defined as the sum of the rectified flow reconstruction loss and the KL divergence loss, with the KL loss weighted at 1.0 for the 1D experiment and 0.1 for the 2D experiment. We employ AdamW as the optimizer with a learning rate of $1 \times 10^{-3}$ and train the two networks $q_\phi$ and $v_\theta$ jointly for 20,000 iterations.

## B  QUALITATIVE RESULTS OF SYNTHETIC 1D EXPERIMENT

We provide qualitative flow visualizations from the synthetic 1D experiment in Fig. 9. Our method effectively captures velocity ambiguity and predicts crossing flows, whereas the baselines produce deterministic outputs.

## C  IMPLEMENTATION DETAILS OF MNIST EXPERTIMENT

In the rectified flow baseline, the velocity network $v_\theta$ uses separate encoders for time $t$ and data $x$. The time $t$ encoder consists of a sinusoidal positional encoding layer followed by two MLP layers with SiLU activation. The data $x$ encoder includes a convolutional in-projection layer, five consecutive ResNet He et al. (2015) blocks (each consisting of two convolutional layers with a kernel size of 3, group normalization, and SiLU activation), followed by a convolutional out-projection layer. The time and data embeddings are concatenated and passed to a decoder composed of a convolutional in-projection layer, five consecutive ResNet blocks, and a convolutional out-projection layer with a kernel size of 1 and an output channel of 1. The hidden dimension is set to 64. MNIST data is normalized to the $[-1, 1]$ range. We adopted the consistency velocity loss from the consistency flow

---

[1]https://github.com/YangLing0818/consistency_flow_matching

matching baseline used for synthetic data experiments. We train the network for 100,000 iterations using the AdamW optimizer with a learning rate of $1 \times 10^{-3}$ and batch size of 256.

In our variational flow matching approach, the velocity network $v_\theta$ includes an additional latent encoding module consisting of a sinusoidal positional encoding layer followed by two MLP layers with SiLU activation. The conditional latent embedding $z$ is concatenated with the embeddings for time $t$ and data $x$. The decoder structure mirrors the baseline, with the first in-projection layer adjusted to handle the increased channel input. The posterior model $q_\phi$ follows a similar architecture, with separate encoders for each input $[x_0, x_1, x_t]$. The resulting embeddings are concatenated and passed through a decoder consisting of a convolutional in-projection layer, followed by three consecutive interleaving ResNet blocks and average pooling layers. The final hidden activation is flattened and processed by two linear MLP layers to predict the 1D latent $z$ with a dimension of 2. The two networks are trained jointly for 100,000 iterations using the AdamW optimizer with a learning rate of $1 \times 10^{-3}$ and a batch size of 256. The KL loss weight is set to $1 \times 10^{-3}$.

# D    IMPLEMENTATION DETAILS OF CIFAR-10 EXPERTIMENT

We directly adopt the rectified flow baseline from Tong et al. (2024) and add modules to incorporate conditional signals from a 1D latent $z$. For both conditioning mechanisms discussed in Section 4.4, the sampled latent is processed through two MLP layers with SiLU activation, with hidden and output dimensions set to 512.

In the adaptive norm variant, the latent embedding $z$ is combined with the time embedding from $v_\theta$ to regress the learnable scale and shift parameters $\gamma$ and $\beta$ for the adaptive group norm layers. For the bottleneck sum variant, the latent is added to the bottleneck feature of $v_\theta$. Since the lowest spatial resolution of the baseline network is $4 \times 4$, the 1D latent is spatially repeated and fused with the bottleneck feature via a weighted sum. To ensure effective use of the latent, we assign a weighting of 0.9 to the latent and 0.1 to the original velocity feature.

The posterior model $q_\phi$ shares a similar encoder structure to $v_\theta$ but omits the decoder. To achieve greater spatial compression, we increase the number of downsampling blocks, predicting features at a $1 \times 1$ spatial resolution. The base channel size is set to 16. Both networks are trained jointly for 600,000 iterations using the Adam optimizer with a learning rate of $2 \times 10^{-4}$ and a batch size of 128. The KL loss weighting is presented alongside the results in Table 1.

# E    ON PRESERVING THE MARGINAL DATA DISTRIBUTION

We obtain samples by numerically solving the ordinary differential equation

$$du_t = v_\theta(x_t, t, z)dt \quad \text{with} \quad z \sim p(z) = \mathcal{N}(z; 0, I).$$

This differs slightly from Theorem 3.3 of Liu et al. (2023) because the velocity $v_\theta$ depends on a latent variable $z$ drawn from a standard Gaussian.

However, Theorem 3.3 of Liu et al. (2023) can be extended to fit this setting as follows.

First, note that we have $v^*(x_t, t, z) = \mathbb{E}[\dot{X}_t | X_t, Z]$ where $X_t$ and $Z$ are random variables corresponding to instances $x_t$ and $z$.

Incorporating the velocity field depending on the latent variable $z$ into the transport problem defined in Eq. (2) and taking an expectation over the latent variable, we obtain the continuity equation

$$\dot{p}_t + \text{div}(\mathbb{E}_Z[v_\theta(x_t, t, z)]p_t) = 0. \tag{6}$$

Following Liu et al. (2023), one can show equivalence to the following equality, which uses any compactly supported continuously differentiable test function $h$:

$$\frac{d}{dt}\mathbb{E}[h(X_t)] = \mathbb{E}[\nabla h(X_t)^T \dot{X}_t] = \mathbb{E}[\nabla h(X_t)^T v^*(X_t, t)] = \mathbb{E}_X[\nabla h(X_t)^T \mathbb{E}_Z[v^*(X_t, t, Z)]].$$

Concretely, equivalence can be shown via

$$0 = \mathbb{E}_Z\left(\int_{x_t} h(\dot{p}_t + \text{div}(v^*(X_t, t, Z)p_t))\right) = \frac{d}{dt}\mathbb{E}[h(X_t)] - \mathbb{E}_X[\nabla h(X_t)^T \mathbb{E}_Z[v^*(X_t, t, Z)]].$$

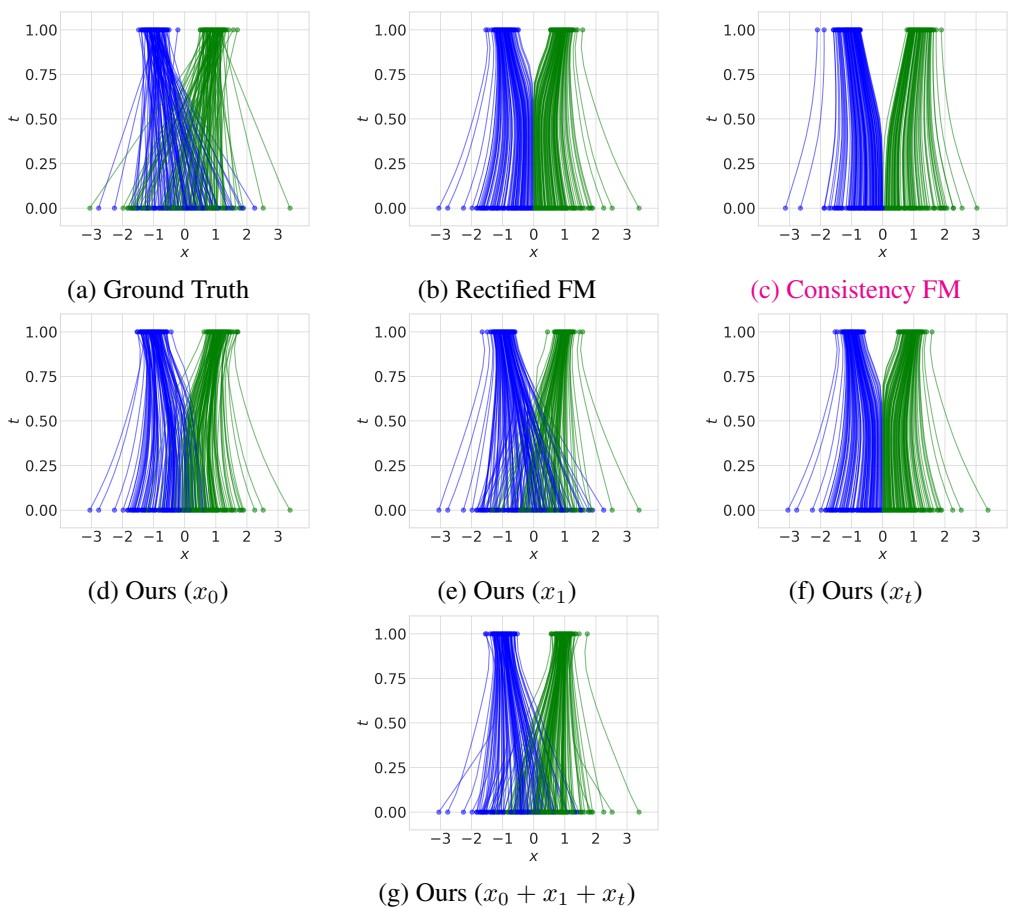

(a) Ground Truth      (b) Rectified FM      (c) Consistency FM

(d) Ours ($x_0$)      (e) Ours ($x_1$)      (f) Ours ($x_t$)

(g) Ours ($x_0 + x_1 + x_t$)

Figure 9: 1D flow visualization for Unimodal Gaussian to Bimodal Gaussian.

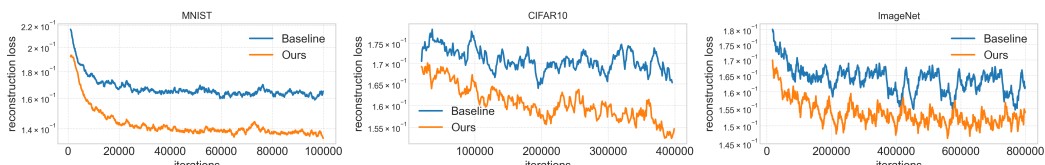

Figure 10: Reconstruction loss for MNIST (left), CIFAR-10 (middle), and ImageNet (right). We observe lower reconstruction losses for the variational formulation, indicating a better fit.

Note, different from Liu et al. (2023), in our case $U_t$ is driven by a velocity field $v(x_t, t, z)$ that depends on a latent variable. Averaging over instantiations of the random latent variable $Z$ leads to the same marginal velocity that appears in the continuity equation (Eq. (6)). Therefore, we solve the same equation with the same initial condition ($X_0 = U_0$). Equivalence follows if the solution to Eq. (6) is unique.

## F    RECONSTRUCTION LOSS VISUALIZATIONS

We present the reconstruction loss curves for our model and the baseline trained on MNIST, CIFAR-10, and ImageNet data in Fig. 10. We observe better reconstruction losses of our model compared to vanilla rectified flow, indicating that the predicted velocities more accurately approximate the ground-truth velocities.

| NFE / sample | 2 | 5 | 10 | 50 | 100 | 1000 | Adaptive |
|---|---|---|---|---|---|---|---|
| I-CFM (Liu et al., 2023; Tong et al., 2024) | 2.786 | 7.143 | 8.326 | 8.770 | 8.872 | 9.022 | 9.041 |
| 1  VRFM (adaptive norm, $x_1$, 2e-3) | 3.943 | 7.728 | 8.499 | 8.973 | 9.050 | 9.168 | 9.171 |
| 2  VRFM (adaptive norm, $x_1$, 5e-3) | 3.083 | 7.202 | 8.342 | 8.868 | 8.997 | 9.166 | 9.183 |
| 3  VRFM (adaptive norm, $x_1 + t$, 5e-3) | **4.460** | **7.930** | **8.583** | **9.007** | **9.104** | **9.220** | **9.238** |

Table 2: Inception Score evaluation of our method compared to the baseline on CIFAR-10, using fixed-step Euler and adaptive-step Dopri5 ODE solvers. Higher scores indicate better performance.

| NFE / sample | 2 | 5 | 10 | 50 | 100 | 1000 | Adaptive |
|---|---|---|---|---|---|---|---|
| OT-FM (Lipman et al., 2023; Tong et al., 2024) | 166.655 | 36.188 | 14.396 | 5.557 | 4.640 | 3.822 | 3.655 |
| I-CFM (Liu et al., 2023; Tong et al., 2024) | 168.654 | 35.489 | 13.788 | **5.288** | 4.461 | 3.643 | 3.659 |
| 1  VRFM-L (100% Posterior Model) | **135.275** | **28.912** | 13.226 | 5.382 | 4.430 | **3.642** | **3.545** |
| 2  VRFM-M (17.5% Posterior Model) | 135.983 | 30.106 | 13.783 | 5.486 | 4.500 | 3.697 | 3.607 |
| 3  VRFM-S (6.7% Posterior Model) | 144.676 | 31.224 | 13.406 | 5.289 | 4.398 | 3.699 | 3.639 |

Table 3: We use the same flow matching model $v_\theta$ and pair it with different sizes of encoders $q_\phi$ during training while maintaining the exact same hyper-parameters. We report the FID scores for our method and the baseline using both fixed-step Euler and adaptive-step Dopri5 ODE solvers.

## G  INCEPTION SCORE EVALUATION

We evaluate the Inception Score of our model trained on CIFAR-10 data and present results in Table 2. This score quantifies the distribution of predicted labels for the generated samples. Compared to the vanilla rectified flow baseline, our method consistently achieves higher Inception Scores, reflecting improved diversity in the generated samples.

## H  ABLATION ON POSTERIOR MODEL SIZE

We conducted ablations to study the impact of varying the size of the encoder $q_\phi$, reducing it to 6.7% and 17.5% of its original size. The results reported in Table 3 demonstrate that our model maintains comparable performance across these variations, highlighting the flexibility and robustness of our approach.

## I  IMAGENET EXPERIMENTS

We conduct experiments on the ImageNet $64 \times 64$ dataset, using the same training setup as we used for the CIFAR-10 experiments, i.e., no additional hyperparameter tuning or cherry-picking. The only changes: increasing the number of iterations to 800k and adjusting the batch size to 128 to accommodate the larger training set. The resulting FID scores, summarized in Table 4, show that our method consistently outperforms the baseline models, even on this larger, real-world dataset. These findings demonstrate the scalability and effectiveness of our approach in handling more complex data while maintaining its advantage over baseline methods.

## J  CONDITIONAL IMAGE GENERATION EXPERIMENT

We also present results for class-conditional generation on the CIFAR-10 and ImageNet dataset. The results are presented in Table 5 and Table 6. Our method consistently outperforms the baseline across different function evaluations.

| NFE / sample | 2 | 5 | 10 | 50 | 100 | 1000 | Adaptive |
|---|---|---|---|---|---|---|---|
| I-CFM (Liu et al., 2023; Tong et al., 2024) | 194.134 | 70.008 | 44.088 | 32.385 | 31.218 | 29.787 | 29.445 |
| VRFM (adaptive norm, $x_1$, 2e-3) | 189.146 | 66.245 | 40.649 | 30.170 | 29.368 | 28.338 | 28.228 |
| VRFM (adaptive norm, $x_1$, 5e-3) | 192.516 | 67.058 | 41.058 | 29.919 | **28.824** | **27.483** | **27.330** |
| VRFM (adaptive norm, $x_1 + t$, 5e-3) | **168.020** | **55.639** | **37.382** | **29.619** | 28.826 | 27.794 | 27.530 |

Table 4: FID Score evaluation of our method compared to the baseline on ImageNet, using fixed-step Euler and adaptive-step Dopri5 ODE solvers. Lower scores indicate better performance.

| NFE / sample | 2 | 5 | 10 | 50 | 100 | 1000 | Adaptive |
|---|---|---|---|---|---|---|---|
| I-CFM (Liu et al., 2023; Tong et al., 2024) | 109.34951 | 23.87121 | 11.817 | 4.787 | 3.858 | 3.107 | 3.046 |
| VRFM (adaptive norm, $x_1$, 2e-3) | 104.708 | 22.677 | **11.380** | **4.391** | **3.539** | **2.869** | **2.824** |
| VRFM (adaptive norm, $x_1 + t$, 5e-3) | **97.341** | **22.245** | 11.580 | 4.552 | 3.638 | 2.910 | 2.853 |

Table 5: FID Score evaluation of class-conditional generation on CIFAR-10, using fixed-step Euler and adaptive-step Dopri5 ODE solvers. Lower scores indicate better performance.

# K    ADDITIONAL RELATED WORK DISCUSSION

Here, we discuss related work aimed at improving the sample efficiency of diffusion and flow matching models, either via consistency modeling or via distillation. We used work by Yang et al. (2024) as the consistency model baseline because it improved upon earlier consistency modeling work by Song et al. (2023); Kim et al. (2023) and also distillation work by Nguyen et al. (2024). Specifically, we used the publicly available baseline.[2]

**Consistency models.** Consistency models, such as those by Song et al. (2023) and Yang et al. (2024), enforce self-consistency across timesteps, ensuring trajectories map back to the same initial point. Moreover, Kim et al. (2023) ensure consistent trajectories for probability flow ODEs. While consistency models focus on improving performance via trajectory alignment if few function evaluations are used, they don't model the multimodal ground-truth distribution, which is our goal.

To illustrate this, we trained the recently developed consistency flow matching model proposed by Yang et al. (2024) (which improves upon work by Song et al. (2023) and Kim et al. (2023); both are not flow matching based) on the data for which VRFM results are presented in Figs. 3 and 9. We obtain the results illustrated in Fig. 11. As expected, we observe that classic consistency modeling does not capture the multimodal velocity distribution, unlike the proposed VRFM.

We also want to note that we think consistency models are orthogonal to our proposed variational formulation. Hence, we think it is exciting future research to study the combination of variational formulations and consistency models, which is beyond the scope of this paper.

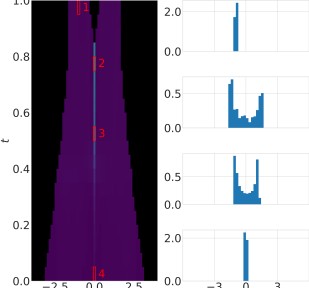

Figure 11: Velocity distribution of consistency flow matching (Yang et al., 2024).

**Distillation.** Nguyen et al. (2024) perform distillation by optimizing step sizes in pretrained flow-matching models to refine trajectories and improve training dynamics. Moreover, Yan et al. (2024) perform distillation by introduceing a piecewise rectified flow mechanism to accelerate flow-based generative models . Note, both methods distill useful information from a pretrained model, either by using dynamic programming to optimize the step size or by applying reflow to straighten trajectories, i.e., they focus on distilling already learned models. In contrast, our VRFM focuses on learning via single-stage training, directly from ground-truth data, and without use of a pre-trained deep net, a flow-matching model, which captures a multimodal velocity distribution.

---

[2]https://github.com/YangLing0818/consistency_flow_matching

| NFE / sample | 2 | 5 | 10 | 50 | 100 | 1000 | Adaptive |
|---|---|---|---|---|---|---|---|
| I-CFM
(Liu et al., 2023; Tong et al., 2024) | 132.139 | 38.421 | 23.614 | 19.078 | 18.611 | 18.088 | 18.066 |
| VRFM (adaptive norm, $x_1$, 2e-3) | **124.718** | **34.453** | **20.632** | **16.408** | **15.999** | **15.440** | **15.521** |
| VRFM (adaptive norm, $x_1 + t$, 5e-3) | 128.773 | 35.848 | 22.186 | 17.579 | 17.090 | 16.541 | 16.567 |

Table 6: FID Score evaluation of class-conditional generation on ImageNet, using fixed-step Euler and adaptive-step Dopri5 ODE solvers. Lower scores indicate better performance.

More research on the distillation of a VRFM model is required to assess how multimodality can be maintained in the second distillation step. We think this is exciting future research, which is beyond the scope of this paper.

## L QUALITATIVE RESULTS OF CIFAR-10 EXPERIMENT

We present qualitative results of our model trained on CIFAR-10 data in Fig. 12.

972
973
974
975
976
977
978
979
980
981
982
983
984
985
986
987
988
989
990
991
992
993
994
995
996
997
998
999
1000
1001
1002
1003
1004
1005
1006
1007
1008
1009
1010
1011
1012
1013
1014
1015
1016
1017
1018
1019
1020
1021
1022
1023
1024
1025

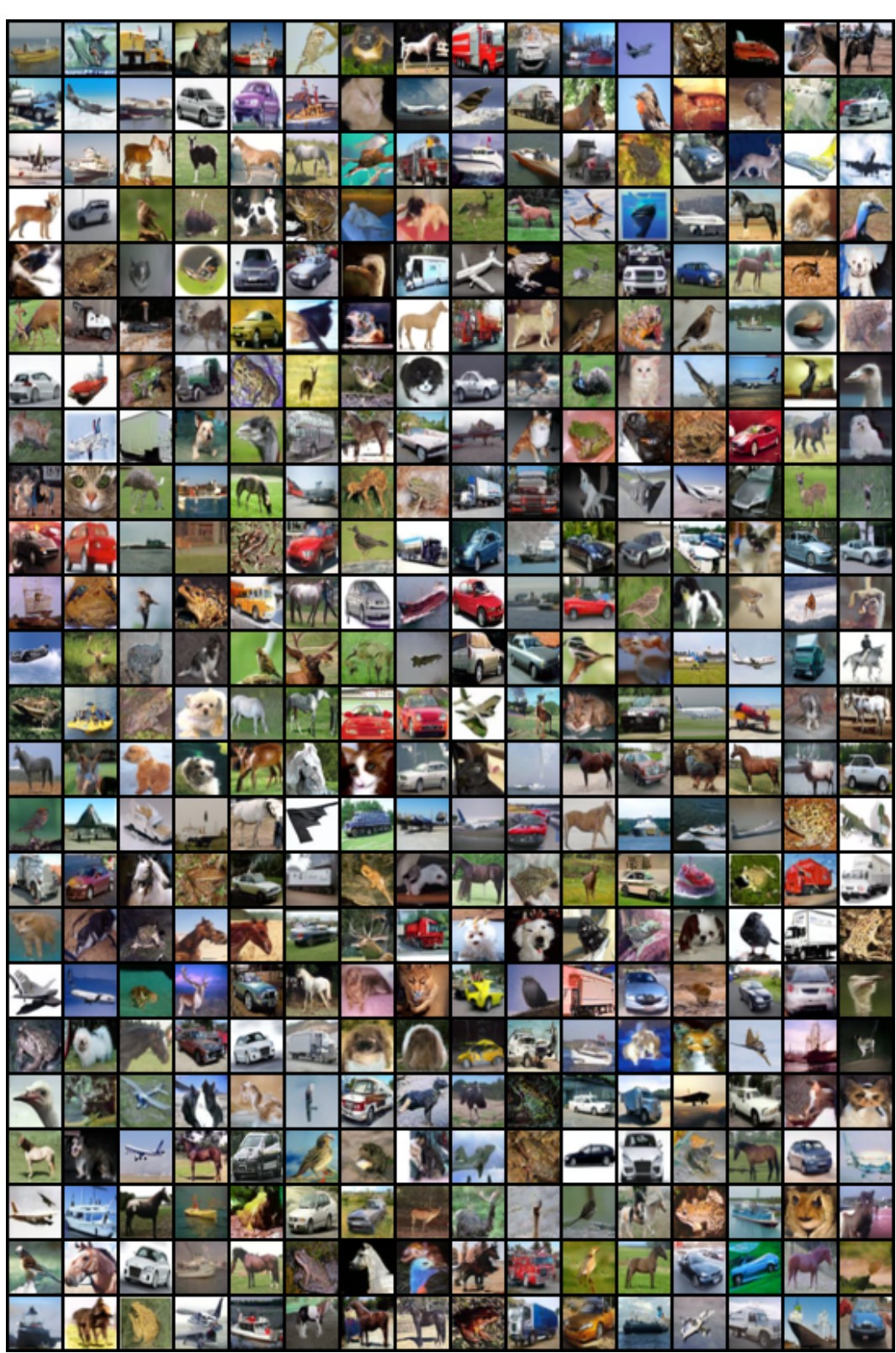

Figure 12: Samples generated from our model trained on CIFAR-10 data.

