# OpenReview forum: "Variational Rectified Flow Matching"
_ICLR.cc/2025/Conference — Submitted to ICLR 2025_

### Official Review · Reviewer_Y61o · 2024-10-28

**Soundness:** 3
**Presentation:** 2
**Contribution:** 3
**Rating:** 6
**Confidence:** 4

**Summary:**

This paper presents Variational Rectified Flow Matching (VRFM), a framework that improves classic rectified flow matching by incorporating multi-modal velocity vector fields based on a variational perspective. Previous flow matching approaches average out directions, leading to curved integration paths and hindering accurately fitting the target distribution. VRFM, by contrast, captures the multi-modality of flow directions, thus preserving directional diversity. Experimental results on synthetic data, MNIST, and CIFAR-10 demonstrate that VRFM achieves promising results with fewer integration steps.

**Strengths:**

- Significance: This paper addresses a notable limitation in flow matching models, specifically the ambiguity at path intersections that results in curved sampling trajectories. By tackling this ambiguity, the proposed approach demonstrates clear improvements over existing rectified flow models, particularly in low NFE settings.

- Originality and clarity: The paper is well-written and easy to follow, clearly presenting concepts. Interpreting the flow-matching objective through variational inference to reduce directional ambiguity is conceptually sound, adding a meaningful perspective to the flow-matching framework.

**Weaknesses:**

- In line 53, the authors claim, “This results in trajectories that are more straight”.  It is unclear why reducing ambiguity would inherently lead to straighter flows. Including a theoretical proof or a detailed explanation to clarify this result would strengthen the argument.

- For completeness, the paper should include proofs demonstrating that the learned distribution from VRFM preserves the marginal data distribution, as established in Theorem 3.3 in [1].

- The most concerning part of this paper is limited evaluation and performance compared to the recent papers. The empirical evaluation is restricted to MNIST and CIFAR-10, which limits the generalizability of the findings. Extending the evaluation to additional datasets, such as ImageNet 64x64, would improve the generalizability of the findings. Furthermore, the reported results of VRFM in low NFE regimes (e.g., 104 FID for 2 NFE on CIFAR-10) are less compelling, given the recent advances [1,2,3] in reducing sampling costs in diffusion (or rectified flow) models. For instance, reflow on rectified flow (e.g., 2-rectified flow) achieves a 4.85 FID with a single step [1]. Results showing VRFM’s performance with the reflow technique would provide a more competitive comparison.

- It would be valuable if the authors could provide results on conditional generation setting.

**Questions:**

- Is there any reason for designing the input of the posterior encoder as [x0, x1, xt, t]?

---

> ### Author Response · Authors · 2024-11-22
> **Response to Reviewer Y61o - Part 1**
>
> Thanks for your time and feedback.
>
> **Reviewer Comment 1:** unclear why reducing ambiguity would inherently lead to straighter flows.
>
> **Response:** We demonstrate that resolving the ambiguity problem allows our model to better match the ground-truth (GT) flow, which is linear by definition in rectified flow. Specifically, a lower loss in Equation (5) indicates that the predicted flow more closely aligns with the GT flow, inherently leading to straighter trajectories.
>
> During training, we observed that our method achieves better velocity reconstruction losses (Appendix F, Figure 10)  compared to vanilla rectified flow, indicating that the predicted velocities more accurately approximate the GT velocities, which are linear. Furthermore, in the experimental section, we provide strong empirical evidence that our method outperforms baseline approaches, especially when the number of function evaluations (NFEs) is small on multiple benchmarks (Synthetic, MNIST, CIFAR-10, ImageNet) in the experiments section and the newly added Appendix G-J. This highlights the practical benefits of our approach in efficiently modeling flows while addressing the velocity ambiguity issue.
>
> **Reviewer Comment 2:** proofs demonstrating that the learned distribution from VRFM preserves the marginal data distribution.
>
> **Response:** We include the derivation in the newly added Appendix E.
>
> **Reviewer Comment 3:** empirical evaluation is restricted to MNIST and CIFAR-10, which limits the generalizability of the findings.
>
> **Response:** To answer, we conduct additional experiments on the ImageNet 64x64 dataset. The training setup and architecture is exactly identical to our CIFAR-10 training, i.e., no additional hyperparameter tuning or cherry-picking. The only changes: increasing the number of iterations to 800k and adjusting the batch size to 128 to accommodate the larger training set. The resulting FID scores are summarized below and added in Appendix I, Table 4. We observe that our method improves upon the baseline, even in this large-scale real-world dataset. These results demonstrate the scalability and effectiveness of our approach in handling more complex data while maintaining its advantages over baseline methods.
>
> | NFE / sample                        | 2                        | 5                       | 10                      | 50                      | 100                     | 1000                    | Adaptive                |
> | ----------------------------------- | ------------------------ | ----------------------- | ----------------------- | ----------------------- | ----------------------- | ----------------------- | ----------------------- |
> | I-CFM                               | 194.134                  | 70.008                  | 44.088                  | 32.385                  | 31.218                  | 29.787                  | 29.445                  |
> | VRFM (adaptive norm, $x_1+t$, 5e-3) | **168.020** | **55.639** | **37.382** | **29.619** | **28.826** | **27.794** | **27.530** |

---

> > ### Author Response · Authors · 2024-11-22
> > **Response to Reviewer Y61o - Part 2**
> >
> > **Reviewer Comment 4:** Results showing VRFM’s performance with the reflow technique would provide a more competitive comparison.
> >
> > **Response:** We believe that the mentioned reflow paper represents an orthogonal direction to our work. Reflow, like other related methods, builds upon classic flow matching. In contrast, VRFM introduces a variational framework that models multimodal velocity distributions, allowing for more flexibility in handling flow ambiguity. For this reason, we consider rectified flow as our primary baseline. Integrating our approach with additional techniques, such as reflow, is beyond the scope of this work.
> >
> > That said, it is technically feasible to integrate reflow into the VFM framework. Specifically, after completing one round of training, we can resample $(x_0,x_1)$ pairs. Unlike classic flow matching, VFM enables us to sample multiple $x_1$ values for a given $x_0$, leveraging the model’s ability to capture multimodal distributions. By consolidating these resampled pairs into a new dataset, we can initiate a subsequent round of VFM training.
> >
> > 1. If ambiguity is modeled nearly perfectly by our variational model in the first round (e.g., for our 1D/2D data), reflow won’t help too much, which is expected as we already attain a very high evaluation score even for very low neural function evaluations (NFE) in those settings.
> > 2. If ambiguity is not modeled perfectly by our variational model in the first round (e.g., for cifar10/imagenet data), reflow will help the model to find more dependent data pairs. As we shown in Figure 8 in the main paper, varying different $z$ will only change the color patterns, while an image’s content is primarily determined by $x_0$. Thus with the new reflow algorithm, we expect to construct an “easier” set for the model to capture, indicating it could be helpful to further improve performance.
> >
> > This combination is an exciting avenue for future work, and we believe that integrating the two methods could lead to advancements in the performance and scalability of flow-based generative models.
> >
> > **Reviewer Comment 5:** results with conditional generation settings.
> >
> > **Response:** Following the reviewer’s suggestion, we extend our experiments to class-conditional generation on the CIFAR-10 dataset. Results are presented below and in the newly added Appendix J, Table 5. Our method consistently outperforms the baseline across different function evaluations.
> >
> > | NFE / sample| 2 | 5 | 10| 50 | 100| 1000 | Adaptive |
> > | --- | --- | --- | --- | --- | --- | --- | --- |
> > | I-CFM | 109.34951 | 23.87121| 11.817| 4.787| 3.858| 3.107| 3.046|
> > | VRFM (adaptive norm, $x_1$, 2e-3) | *104.708*| *22.677* | **11.380** | **4.391** | **3.539** | **2.869** | **2.824** |
> > | VRFM (adaptive norm, $x_1+t$, 5e-3) | **97.341** | **22.245** | *11.580* | *4.552* | *3.638* | *2.910* | *2.853* |
> >
> > **Reviewer Comment 6:**  Is there any reason for designing the input of the posterior encoder as $[x0, x1, xt, t]$?
> >
> > **Response:** Since the variational rectified flow model is conditioned on a latent variable sampled from a prior distribution, any combination of these arguments can, in principle, be used to construct the posterior distribution. As discussed in Section 4.1 (Lines 317–323 and Figure 3), we observe an intriguing relationship between the ability to model velocity ambiguity at different timesteps $t$ and the choice of posterior input. Specifically:
> >
> > * When conditioning only on $x_0$, the model struggles to predict a bi-modal distribution at early timesteps due to the lack of information about $x_1$.
> > * When conditioning on $x_1$, the model fails to capture the ground-truth distribution at later timesteps as the influence of $x_1$ diminishes over time.
> > * When conditioning on $x_t$, the ambiguity plot resembles that of the baselines, as no additional information beyond the current flow state is provided.
> >
> > These findings provide an empirical analysis of how different conditioning signals affect the model's ability to capture velocity ambiguity. The choice of $[x0,x1,xt,t]$ as input was made to balance the information provided across timesteps and address the limitations observed when using subsets of these arguments.
> > That said, we believe there is exciting future work to further explore the theoretical and empirical connections between posterior encoder inputs and the resulting model performance across diverse tasks. Such an investigation could yield deeper insights into the optimal design of the posterior encoder.

---

> ### Comment · Reviewer_Y61o · 2024-11-25
>
> Thank you for providing the detailed explanations, proofs, and additional experiments. I appreciate the effort to address my initial concerns, and I find the variational perspective introduced for rectified flow both novel and interesting.
>
> That said, I believe there are still limitations in the practical performance of VRFM, particularly at small NFEs. The additional ImageNet experiments, while helpful, reveal an FID score that remains noticeably behind state-of-the-art models (e.g., FID below 3). Even at larger NFEs (e.g., 1000), the performance does not reach the performance of strong baselines. Given that VRFM’s primary contribution is in reducing ambiguities at intersections and producing straighter flows, I anticipated a more substantial improvement that could bridge the gap with state-of-the-art methods.
>
> In conclusion, while I find the proposed direction intriguing and promising, the current approach seems to require further advancements to meet the standard of a top-tier conference like ICLR. Based on the improvements provided in the rebuttal and the potential of this direction, I am increasing my rating to 6.

---

> > ### Author Response · Authors · 2024-11-26
> >
> > **Reviewer comment**: ImageNet FID score behind state-of-the-art models.
> >
> > **Response**:
> >
> > First, we think the reported FID values are reasonable. The performance difference is due to model size differences and generation settings. For instance, OT-FM [1] employs significantly larger models (296M parameters) to achieve a 14.45 FID in unconditional generation. In contrast, our model with only 37.65M parameters achieves a 27.53 FID. Additionally, in new experiments, we trained and evaluated VRFM in class-conditional generation on ImageNet-64, achieving an FID of 15.521 (as shown in the Table below), which outperforms the rectified flow baseline by a noticeable margin. For further context, SiT-S [2], operating with a comparable model size (33M parameters), achieves a 57.6 FID in class-conditional generation at the more challenging 256x256 resolution. Considering these differences, we believe our results are reasonable given the network size and experimental setup.
> >
> > Second, our objective is to ensure a maximally fair evaluation. To achieve this, we employ the same velocity network, $v_\theta$, and train it using both vanilla rectified flow matching (I-CFM) and our proposed variational rectified flow matching (VRFM), based on a publicly available codebase [3]. We assess performance on CIFAR-10, presenting results for unconditional generation in Table 1 of the main text and for conditional generation in Table 5 of the appendix. Additionally, we provide an updated ImageNet table below, covering both unconditional and conditional generation. These results are also added to the Appendix as Tables 4 and 6. Across these comprehensive experimental settings, we consistently observe that VRFM outperforms the baseline, demonstrating its effectiveness.
> >
> > Table: unconditional generation for ImageNet
> > | NFE / sample| 2| 5 | 10| 50 | 100| 1000 | Adaptive |
> > | --- | --- | --- | --- | --- | --- | --- | --- |
> > | I-CFM | 194.134 | 70.008 | 44.088 | 32.385 | 31.218 | 29.787 | 29.445 |
> > | VRFM (adaptive norm, $x_1$, 2e-3) | *189.146* | *66.245* | *40.649* | *30.170* | *29.368* | *28.338* | *28.228* |
> > | VRFM (adaptive norm, $x_1+t$, 5e-3) | **168.020** | **55.639** | **37.382** | **29.619** | **28.826** | **27.794** | **27.530** |
> >
> > Table: conditional generation for ImageNet
> > | NFE / sample| 2| 5 | 10| 50 | 100| 1000 | Adaptive |
> > | --- | --- | --- | --- | --- | --- | --- | --- |
> > | I-CFM | 132.139 | 38.421 | 23.614 | 19.078 | 18.611 | 18.088 | 18.066 |
> > | VRFM (adaptive norm, $x_1$, 2e-3) | **124.718** | **34.453** | **20.632** | **16.408** | **15.999** | **15.440** | **15.521** |
> > | VRFM (adaptive norm, $x_1+t$, 5e-3) | *128.773* | *35.848* | *22.186* | *17.579* | *17.090* | *16.541* | *16.567* |
> >
> > Importantly, similar to prior work [3], we want to emphasize that the primary goal of this paper is not to set a new SOTA benchmark, but to study a novel aspect of flow-based models: ambiguity/multimodality in velocity fields. By introducing a variational formulation that explicitly models the multimodal velocity distribution and enables intersecting flows, our work explores an angle that has been overlooked by prior work. We validate our contribution through extensive experiments across a range of settings, from controlled 1D/2D tasks to MNIST, CIFAR-10, and ImageNet data.
> >
> > We hope to 1) introduce a new direction for research, and to 2) encourage the research community to enhance our formulation by scaling up model sizes and integrating advanced training technics (i.e., noise schedule, training hyper-parameters, transformer architectures, etc.) to further improve performance.
> >
> > [1] Y. Lipman, R. Chen, H. Ben-Hamu, M. Nickel, and M. Le. Flow Matching for Generative Modeling. In Proc. ICLR, 2023.
> >
> > [2] Ma, N., Goldstein, M., Albergo, M.S., Boffi, N.M., Vanden-Eijnden, E. and Xie, S., 2024. Sit: Exploring flow and diffusion-based generative models with scalable interpolant transformers. arXiv preprint arXiv:2401.08740.
> >
> > [3] Tong, A., FATRAS, K., Malkin, N., Huguet, G., Zhang, Y., Rector-Brooks, J., Wolf, G. and Bengio, Y., Improving and generalizing flow-based generative models with minibatch optimal transport. Transactions on Machine Learning Research.

---

> > > ### Author Response · Authors · 2024-12-02
> > >
> > > Thank you for taking the time to share your valuable feedback. We hope our responses and revisions answered your questions. Please reach out with any additional questions you may have. We look forward to hearing from you.

---

> ### Comment · Reviewer_Y61o · 2024-12-03
>
> I acknowledge that VRFM improves vanilla rectified flow under comparable model sizes, and I appreciate the efforts to highlight this advancement. However, I maintain my current score for two primary reasons:
>
> 1) The results remain unconvincing, particularly at low NFEs. Given that the proposed method argues that it reduces ambiguity and learns straighter flows, I expect a more substantial performance gain in low NFE regimes. The significant performance gap between low and high NFEs falls short of these expectations.
>
> 2) The main claim of this paper is improved performance in accelerated settings, as mentioned in the abstract and introduction section. To substantiate this claim, I believe it is necessary to include comparisons against other baselines that also target improvements in accelerated settings. Additionally, I believe it is within scope to apply the reflow procedure to VRFM, as it aligns with the goal of extending the capabilities of rectified flow.
>
> In conclusion, while I appreciate the contributions of this work, these considerations prevent me from adjusting my score further at this stage.

---

> ### Author Response · Authors · 2024-12-04
>
> We thank the reviewer for detailed feedback and for acknowledging the improvements VRFM achieves over vanilla rectified flow under comparable model sizes. Below, we answer the remaining comments:
>
> **Reviewer comment 1**: significant performance gap between low and high NFEs
>
> As stated in the abstract (L21), our goal is to capture multimodal velocity vector fields in flow matching. We show that the proposed formulation is able to achieve this goal. We further demonstrate that capturing a multimodal velocity vector field leads to consistent improvements over classic flow matching across all NFE regimes (as shown in Tables and Figures in the main paper). Note, we don’t intend to reduce the performance gap between low and high NFEs. Instead, we aim for improvements across all NFE regimes, which the proposed method achieves.
>
> **Reviewer comment 2**: main claim of this paper is improved performance in accelerated settings
>
> As stated in the abstract (L21) and the introduction (L49), our (main) claim is to model the multimodality/ambiguity of the velocity field in flow matching. The empirical results (Figures 1, 3, 10) demonstrate that we successfully achieve this goal. We also observe our method to attain compelling results compared to classic rectified flow across all NFE regimes (including low NFEs) and across datasets including synthetic, MNIST, CIFAR-10, and ImageNet.
>
> **Reviewer comment 3**: comparisons against accelerated baselines and apply reflow to VRFM
>
> We appreciate the reviewer’s suggestion to include comparisons against other accelerated baselines. In our experiments, we included consistency flow matching as a baseline (which itself improves upon recent methods) and found that VRFM outperforms it on synthetic datasets (Figure 2, 5), MNIST when the NFE exceeds 2 (Figure 7), and CIFAR-10 when NFEs exceed 5 (General Response).
>
> It’s a great suggestion to apply reflow to VRFM. As noted in our response to another reviewer (https://openreview.net/forum?id=1cM0yQe3pO&noteId=ZvIn52hRIG), reflow is indeed compatible with our framework and we expect further improvements. However, while reflow is a promising extension, numerous other enhancements—such as advanced noise schedules, alternative regularization strategies, or hybrid generative modeling approaches—could similarly improve VRFM’s capabilities. We think including these extensions dilutes the focus of our core contribution and extends the scope beyond what is practical.
>
> The primary objective of this paper is to introduce VRFM and establish its benefits and shortcomings over classic rectified flow across a variety of settings. While we recognize the potential of other advances to further improve performance, we believe that it is more appropriate to focus on the foundational contribution of VRFM and leave additional integrations to future exploration by the community.

---

### Official Review · Reviewer_obZd · 2024-11-02

**Soundness:** 2
**Presentation:** 2
**Contribution:** 1
**Rating:** 3
**Confidence:** 5

**Summary:**

The paper introduces Variational Rectified Flow Matching as a method to model multi-modal velocity and ambiguity in the data-space-time-space domain. The properties of Variational Rectified Flow Matching are studied and validated through experiments with visualizations on low-dimensional synthetic data. Compelling results are demonstrated on the synthetic data, MNIST, and CIFAR-10 datasets.

**Strengths:**

1. The paper presents a valuable observation: because the vector field is parameterized via a Gaussian at each data-domain-time-domain location, ambiguity cannot be captured.

2. The analytical experiments with visualizations are well-executed and contribute significantly to validating the theoretical analysis.

**Weaknesses:**

1. Current evaluations are too weak and this paper lacks enough experiments on more real-world and complex datasets, such as the AFHQ, CelebA and ImageNet datasets. The evaluations on these benchmarks are necessary for demonstrating the effectiveness of the proposed method.

2. The authors should sufficiently discuss these missing related works [1][2][3][4][5], and compare with them in the experiments.

[1] Nguyen B, Nguyen B, Nguyen V A. Bellman Optimal Stepsize Straightening of Flow-Matching Models[C]//The Twelfth International Conference on Learning Representations. 2024.

[2] Song, Yang, et al. Consistency models. arXiv preprint arXiv:2303.01469 (2023).

[3] Yang, Ling, et al. Consistency flow matching: Defining straight flows with velocity consistency. arXiv preprint arXiv:2407.02398 (2024).

[4] Yan, Hanshu, et al. Perflow: Piecewise rectified flow as universal plug-and-play accelerator. arXiv preprint arXiv:2405.07510 (2024).

[5] Kim, Dongjun, et al. Consistency trajectory models: Learning probability flow ode trajectory of diffusion. arXiv preprint arXiv:2310.02279 (2023).

**Questions:**

I am curious whether Variational Rectified Flow Matching can enhance performance through "reflow", similar to classic rectified flow matching as discussed in [1].


[1] Liu, Xingchao, and Chengyue Gong. "Flow Straight and Fast: Learning to Generate and Transfer Data with Rectified Flow." The Eleventh International Conference on Learning Representations.

---

> ### Author Response · Authors · 2024-11-22
> **Response to Reviewer obZd - Part 1**
>
> Thanks for your time and feedback.
>
> **Reviewer Comment 1:** lacks enough experiments on more real-world and complex datasets like ImageNet.
>
> **Response:** To answer, we conduct additional experiments on the ImageNet 64x64 dataset. The training setup and architecture is exactly identical to our CIFAR-10 training, i.e., no additional hyperparameter tuning or cherry-picking. The only changes: increasing the number of iterations to 800k and adjusting the batch size to 128 to accommodate the larger training set. The resulting FID scores are summarized below and in the newly added Appendix I, Table 4. We observe that the same trends hold for our method compared to the baseline models, even in this large-scale real-world dataset. These results demonstrate the scalability and effectiveness of our approach in handling more complex data while maintaining its advantage over baseline methods.
>
> | NFE / sample                        | 2                        | 5                       | 10                      | 50                      | 100                     | 1000                    | Adaptive                |
> | ----------------------------------- | ------------------------ | ----------------------- | ----------------------- | ----------------------- | ----------------------- | ----------------------- | ----------------------- |
> | I-CFM                               | 194.134                  | 70.008                  | 44.088                  | 32.385                  | 31.218                  | 29.787                  | 29.445                  |
> | VRFM (adaptive norm, $x_1+t$, 5e-3) | **168.020** | **55.639** | **37.382** | **29.619** | **28.826** | **27.794** | **27.530** |
>
> **Reviewer Comment 2:** whether Variational Flow Matching can enhance performance through “reflow”.
>
> **Response:** Great question. Technically, it is feasible to integrate reflow into the VFM framework. Specifically, after completing one round of training, we can resample $(x_0,x_1)$ pairs. Unlike classic flow matching, VFM enables us to sample multiple $x_1$ values for a given $x_0$, leveraging the model’s ability to capture multimodal distributions. By consolidating these resampled pairs into a new dataset, we can initiate a subsequent round of VFM training.
>
> 1. If ambiguity is modeled nearly perfectly by our variational model in the first round (e.g., for our 1D/2D data), reflow won’t help too much, which is expected as we already attain a very high evaluation score even for very low neural function evaluations (NFE) in those settings.
> 2. If ambiguity is not modeled perfectly by our variational model in the first round (e.g., for cifar10/imagenet data), reflow will help the model to find more dependent data pairs. As we shown in Figure 8 in the main paper, varying different $z$ will only change the color patterns, while an image’s content is primarily determined by $x_0$. Thus with the new reflow algorithm, we expect to construct an “easier” set for the model to capture, indicating it could be helpful to further improve performance.
>
> This combination is an exciting avenue for future work, and we believe that integrating the two methods could lead to advancements in the performance and scalability of flow-based generative models.

---

> > ### Author Response · Authors · 2024-11-22
> > **Response to Reviewer obZd - Part 2**
> >
> > **Reviewer Comment 3:** missing discussion and comparison to related works [1,2,3,4,5].
> >
> > **Response:** We thank the reviewer for pointing out the relevant related works. We agree that all the mentioned papers are relevant to our study and represent orthogonal directions to our approach. Below and in the newly added Appendix K, we discuss these works and highlight the key differences:
> >
> > Consistency models, such as those by [1] and [2], enforce self-consistency across timesteps, ensuring trajectories map back to the same initial point. This improves robustness and sample quality in noisy conditions. While effective, these methods focus on deterministic flows and do not explicitly address ambiguity in the ground-truth velocity field. Similarly, [3] ensure consistent trajectories for probability flow ODEs but do not model overlapping or multimodal velocity distributions. In contrast, our Variational Rectified Flow Matching (VRFM) introduces a variational framework to directly address velocity ambiguity by modeling multimodal velocity distributions and enabling intersecting flows. While consistency models focus on improving performance via trajectory alignment if few function evaluations are used, we intend to model the ground-truth velocity as accurately as possible. Importantly, these approaches are orthogonal and complementary—consistency models can be applied to our VRFM framework to enforce trajectory consistency while preserving the flexibility of our model in handling ambiguity. Exploring this integration represents an exciting avenue for future work.
> >
> > Additionally, [4] optimize step sizes in pretrained flow-matching models to refine trajectories and improve training dynamics through distillation, while [5] introduces a piecewise rectified flow mechanism to accelerate flow-based generative models via distillation. While both methods effectively distill useful information from a pretrained model, either by using dynamic programming to optimize the step size or by applying reflow to straighten trajectories, they focus on enhancing already learned models. In contrast, our VRFM focuses on learning a robust flow-matching model directly from ground-truth data. This approach allows us to avoid the information loss inherent in distillation and ensures that these methods can be applied to our learned model to further improve sample efficiency.
> >
> > We see these combinations as promising directions for future research to further improve the capabilities of flow-based models. Their integration is beyond the scope of this paper.
> >
> > [1] Yang Song, Prafulla Dhariwal, Mark Chen, and Ilya Sutskever. Consistency models. In International Conference on Machine Learning, pp. 32211–32252. PMLR, 2023.
> >
> > [2] Ling Yang, Zixiang Zhang, Zhilong Zhang, Xingchao Liu, Minkai Xu, Wentao Zhang, Chenlin Meng, Stefano Ermon, and Bin Cui. Consistency flow matching: Defining straight flows with velocity consistency. arXiv preprint arXiv:2407.02398, 2024.
> >
> > [3] Dongjun Kim, AI Sony, Chieh-Hsin Lai, Wei-Hsiang Liao, Naoki Murata, Yuhta Takida, Yutong He, Yuki Mitsufuji, and Stefano Ermon. Consistency trajectory models: Learning probability flow ode trajectory of diffusion. In Proc. NeurIPS, 2023.
> >
> > [4] Bao Nguyen, Binh Nguyen, and Viet Anh Nguyen. Bellman optimal stepsize straightening of flow-matching models. In The Twelfth International Conference on Learning Representations, 2024.
> >
> > [5] Hanshu Yan, Xingchao Liu, Jiachun Pan, Jun Hao Liew, Qiang Liu, and Jiashi Feng. Perflow: Piecewise rectified flow as universal plug-and-play accelerator. arXiv preprint arXiv:2405.07510, 2024.

---

> > > ### Author Response · Authors · 2024-11-26
> > >
> > > Thank you for taking the time to provide valuable feedback. We hope our responses and revisions have addressed your questions. We look forward to getting to know any further feedback and questions you may have.

---

> > > > ### Comment · Reviewer_obZd · 2024-11-27
> > > >
> > > > Thank the authors for the response. In the rebuttal, the authors do not provide sufficient quantitative comparisons with previous methods [1,2,3,4,5] as I requested, which is critical and necessary. Thus I think current version of paper is not well prepared for publication and I recommend rejection.
> > > >
> > > > [1] Nguyen B, Nguyen B, Nguyen V A. Bellman Optimal Stepsize Straightening of Flow-Matching Models[C]//The Twelfth International Conference on Learning Representations. 2024.
> > > >
> > > > [2] Song, Yang, et al. Consistency models. arXiv preprint arXiv:2303.01469 (2023).
> > > >
> > > > [3] Yang, Ling, et al. Consistency flow matching: Defining straight flows with velocity consistency. arXiv preprint arXiv:2407.02398 (2024).
> > > >
> > > > [4] Yan, Hanshu, et al. Perflow: Piecewise rectified flow as universal plug-and-play accelerator. arXiv preprint arXiv:2405.07510 (2024).
> > > >
> > > > [5] Kim, Dongjun, et al. Consistency trajectory models: Learning probability flow ode trajectory of diffusion. arXiv preprint arXiv:2310.02279 (2023).

---

> > > > > ### Author Response · Authors · 2024-11-28
> > > > >
> > > > > **Reviewer Comment**: Comparison to related works [1,2,3,4,5].
> > > > >
> > > > > **Response**: We sincerely thank the reviewer for the valuable feedback and the active participation in the discussion. Based on the feedback, we added quantitative and qualitative comparisons to consistency models, using as an additional baseline the most recent consistency flow matching (Consistency FM) work by Yang et al. (2024) [3], which improves upon prior consistency model work [2, 5] and distillation [1]. We used the publicly available source code to conduct all experiments.
> > > > >
> > > > > Concretely, we modified the following to include the new results:
> > > > > 1) Fig. 2 to include Consistency FM as a baseline for 1D synthetic data. We observe the proposed approach to maintain an edge across all metrics and evaluation steps, except according to the Wasserstein distance metric if 2 evaluation steps are used.
> > > > > 2) Fig. 4 to include trajectories for Consistency FM for 2D synthetic data. We observe the expected straight line behavior of consistency models.
> > > > > 3) Fig. 5 to include Consistency FM as a baseline for 2D synthetic data. We observe the proposed approach to maintain an edge across all metrics and evaluation steps.
> > > > > 4) Fig. 7 to include Consistency FM as a baseline for MNIST data. We observe the proposed approach to improve results compared to all baselines if the number of evaluation steps is larger than 2. For 2 evaluation steps Consistency FM performs best.
> > > > > 5) Fig. 9 in the appendix to include the trajectories of Consistency FM on 1D synthetic data. We observe the expected straight line behavior of consistency models.
> > > > > 6) The related work section in the main paper to cite work discussing consistency models.
> > > > > 7) Appendix A and Appendix C to describe the implementation details of the Consistency FM baseline.
> > > > > 8) Appendix K to include a plot showing the unimodal velocity distribution of Consistency FM. This plot corroborates our claim that consistency models don’t capture multimodal velocity distributions, which is the goal of our proposed approach.
> > > > >
> > > > > We hope these new results in the latest revision include the desired comparisons and encourage the reviewer to re-consider the recent rating adjustment from an initial “marginally below the acceptance threshold” to a “reject” recommendation. We also think it is exciting future research to combine consistency models and our proposed modeling of multimodal velocity fields. Thanks for the great suggestion.
> > > > >
> > > > > Again, thanks a lot for your time and consideration, and for your active participation in the discussion.

---

> > > > > > ### Author Response · Authors · 2024-12-02
> > > > > >
> > > > > > Thank you for taking the time to share your valuable feedback. We hope our responses and revisions answered your questions. Please reach out with any additional questions you may have. We look forward to hearing from you.

---

### Official Review · Reviewer_qkA9 · 2024-11-03

**Soundness:** 2
**Presentation:** 3
**Contribution:** 2
**Rating:** 6
**Confidence:** 3

**Summary:**

The paper introduces a novel framework, Variational Rectified Flow Matching, which addresses the limitations of conventional Rectified Flow (RF) methods in capturing ambiguity in velocity distributions. By incorporating an additional latent variable z drawn from a Gaussian prior, the framework models multiple modes of ambiguity. An encoder is used to derive the posterior distribution p(v∣xt,t,z) at a specific sample xt and time t. This approach is claimed to better capture ambiguity and improve the empirical performance of diffusion model generation.

**Strengths:**

The strengths of the paper are as follows:

- Clear, easy-to-follow presentation with strong empirical performance compared to baseline methods.

**Weaknesses:**

The weaknesses of the paper are listed below:
- The motivation of the paper is unclear, particularly in the Introduction. The statement at line 73, “Importantly, variational rectified flow matching differs in that it enables modeling ambiguity in the data-space-time-space domain, i.e., the goal is a model where flow trajectories can intersect,” does not clarify how allowing flow trajectories to intersect resolves the ambiguity problem.
- As I understand, using an additional latent variable z when modeling the velocity at (xt,t) can partially address the ambiguity problem, as different values of z capture different modes or sources of variation in the velocity distribution. However, VAEs are known to sometimes experience mode collapse, where the same high-density velocities may be generated from multiple modes of z. How does the proposed method handle this issue? To further address this concern, I suggest including a “diversity metric” in the experimental protocol to measure the variety of generated samples.
- Furthermore, I believe RF can address the ambiguity problem by performing multiple rectifications. When training stabilizes, the optimal velocity at each sample xt at time t becomes unique, eliminating ambiguity. Even without rectification, existing methods such as OT-FM can mitigate ambiguity by improving the coupling between x0 and x1, resulting in less ambiguous directions at (xt, t). Are there any theoretical or methodological benefits of the proposed approach compared to these methods? Without such justification, it’s difficult to attribute the improved performance to the additional variable z.
- Can different values of z affect the visual quality of the generated samples? If x0 is kept constant, does varying z introduce significant variance in the generated samples? Or is there a specific value of z that results in low-quality samples?
- There is no theoretical guarantee that the proposed approach will achieve a better straight flow when addressing the ambiguity problem.

**Questions:**

Refer to the Weaknesses section for my concerns and questions.

---

> ### Author Response · Authors · 2024-11-22
> **Response to Reviewer qkA9 - Part 1**
>
> Thanks for your time and feedback.
>
> **Reviewer Comment 1:** The paper does not clarify how allowing flow trajectories to intersect resolves the ambiguity problem.
>
> **Response:** As discussed in the Introduction (Lines 41–48) and as illustrated in Figure 1, we note that crossing flows point in different directions at the same location in the data-space-time-space domain. Intuitively, being at the same location and moving in different directions means that flows are __crossing__. This phenomenon is also observed for the ground-truth velocities. Hence, at those locations, the (ground-truth) flow trajectories point in multiple directions, i.e., they are ambiguous/multimodal.
>
> Classic flow matching methods are unable to capture this type of ambiguity, as they employ a deterministic squared-norm loss that restricts the model to a single velocity vector at any given point.
>
> In contrast, variational rectified flow matching models the multimodal velocity distribution at any point along the flow in the data-space-time-space domain. By allowing different flow directions at the same data-space-time-space point, our approach enables trajectories to intersect and effectively captures the underlying ambiguity.
>
> We revised Line 72 in the paper to clarify this motivation by stating: “Importantly, variational rectified flow matching differs in that it enables to also model ambiguity in the data-space-time-space domain. This enables different flow directions at the same data-space-time-space point, allowing the resulting flows to intersect at that location. ”
>
> **Reviewer Comment 2:** VAEs are known to sometimes experience mode collapse.
>
> **Response:** While mode collapse is a well-known issue in GANs, where multiple modes of the data distribution may be ignored, VAEs generally do not exhibit the same degree of collapse due to their explicit probabilistic modeling of the latent space. This probabilistic framework inherently encourages coverage of the data distribution's diversity.
> Furthermore, as a standard quantitative metric in generative modeling [1,2,3], the Fréchet Inception Distance (FID) score evaluates both the mean and covariance statistics of the generated data compared to the ground-truth. Sample diversity directly impacts the covariance component of the FID score calculation, making it sensitive to issues like mode collapse or a lack of diversity.
>
> To further address the reviewer’s question, we added the inception score to our evaluation. The score explicitly measures the distribution of predicted labels of the generated samples. Compared to the vanilla rectified flow baseline, our method consistently achieves higher Inception Scores, reflecting improved diversity in the generated samples. By reporting both FID and inception scores, we provide a comprehensive assessment of the model's ability to generate diverse and realistic samples. The full table is available in the new Appendix G, Table 2.
>
> | NFE / sample                          | 2                      | 5                      | 10                     | 50                     | 100                    | 1000                   | Adaptive               |
> | ------------------------------------- | ---------------------- | ---------------------- | ---------------------- | ---------------------- | ---------------------- | ---------------------- | ---------------------- |
> | I-CFM                                 | 2.786                  | 7.143                  | 8.326                  | 8.770                  | 8.872                  | 9.022                  | 9.041                  |
> | VRFM (adaptive norm, $x_1$, 2e-3)     | *3.943*       | *7.728*       | *8.499*       | *8.973*       | *9.050*       | *9.168*       | 9.171                  |
> | VRFM (adaptive norm, $x_1$, 5e-3)     | 3.083                  | 7.202                  | 8.342                  | 8.868                  | 8.997                  | 9.166                  | *9.183*       |
> | VRFM (adaptive norm, $x_1 + t$, 5e-3) | **4.460** | **7.930** | **8.583** | **9.007** | **9.104** | **9.220** | **9.238** |
>
> [1] Tong, A., FATRAS, K., Malkin, N., Huguet, G., Zhang, Y., Rector-Brooks, J., Wolf, G. and Bengio, Y., Improving and generalizing flow-based generative models with minibatch optimal transport. Transactions on Machine Learning Research.
>
> [2] Peebles, W. and Xie, S., 2023. Scalable diffusion models with transformers. In Proceedings of the IEEE/CVF International Conference on Computer Vision (pp. 4195-4205).
>
> [3] Ma, N., Goldstein, M., Albergo, M.S., Boffi, N.M., Vanden-Eijnden, E. and Xie, S., 2024. Sit: Exploring flow and diffusion-based generative models with scalable interpolant transformers. arXiv preprint arXiv:2401.08740.

---

> > ### Author Response · Authors · 2024-11-22
> > **Response to Reviewer qkA9 - Part 2**
> >
> > **Reviewer Comment 3:** RF can address the ambiguity problem by performing multiple rectifications. The optimal velocity at each sample $x_t$ at time $t$ becomes unique. Theoretical or methodological benefits of the proposed approach compared to methods like OT-FM that mitigate ambiguity between $x_0$ and $x_1$.
> >
> > **Response:** Thanks for highlighting ReFlow and OT-FM, which aim to learn straighter flows. We emphasize that these methods do not address the ambiguity of flows. Specifically, these methods are based on the traditional flow matching framework and use a mean-squared-error objective, resulting in a flow model that predicts only a single average velocity at any given location in the data-space-time-space domain.
> >
> > Importantly, in classic flow matching, ambiguity is inherent because samples are matched by randomly drawing from the source and target distributions, as discussed in Section 1 (Lines 39–48). Note further, the learned velocity at each data-space-time-space location $(x_t,t)$ is determined by the training objective which ReFlow and OT-FM don’t modify.
> >
> > In contrast, our method directly tackles this ambiguity by modifying the objective, i.e., our goal is to learn multimodal flow distributions rather than avoiding them. This enables us to model a multimodal velocity distribution at each $(x_t,t)$, enabling the representation of multiple valid flow directions at a single point. This resolves the limitations of objectives in prior works, and captures the inherent uncertainty in the flow.
> >
> > We want to also highlight that our approach is orthogonal to methods like OT-FM. The flexibility of our variational rectified flow framework allows it to be combined with such approaches, offering an exciting direction for future work to further enhance the performance and robustness of our method.
> >
> > **Reviewer Comment 4:** The impact of $z$ on generated samples. Is there a specific value of $z$ that results in low-quality samples?
> >
> > **Response:** As demonstrated in our experiments section, we studied the role of $z$ in both the MNIST dataset (Figure 7) and the CIFAR-10 dataset (Figure 8). As pointed out in Section 4 (Lines 491–496), we observed clear patterns in the generated samples based on $z$. Specifically, images conditioned on the same latent $z$ exhibit consistent color patterns, while images at the same grid location display similar content.
> >
> > Furthermore, since the prior distribution for $z$ is defined as a continuous Gaussian, we did not observe noticeable low-quality samples when $z$ is sampled within meaningful ranges following the prior distribution. However, as is typical for generative models, drawing $z$ completely outside the prior distribution will result in low-quality or nonsensical samples.
> >
> > **Reviewer Comment 5:** theoretical guarantee between addressing velocity ambiguity and resulting in straight flows
> >
> > **Response:** In the paper we demonstrate that resolving the ambiguity problem allows our model to better match the ground-truth (GT) flow (Section 3.1), which is linear by definition in rectified flow. Specifically, a lower reconstruction term in Equation (5) indicates that the predicted flow more closely aligns with the GT flow, inherently leading to straighter trajectories. Quantitatively, we observe better reconstruction losses in our model compared to vanilla rectified flow (Appendix F, Figure 10), indicating that the predicted velocities more accurately approximate the GT velocities, which are linear. Furthermore, in the experimental section, we provide empirical evidence that our method outperforms baseline approaches in multiple benchmarks (experiment section, newly added Appendix G-J), especially when the number of function evaluations (NFEs) is small. This highlights the practical benefits of our approach in efficiently modeling flows while addressing the velocity ambiguity.

---

> > > ### Author Response · Authors · 2024-11-26
> > >
> > > Thank you for taking the time to provide valuable feedback. We hope our responses and revisions have addressed your questions. We look forward to getting to know any further feedback and questions you may have.

---

> ### Comment · Reviewer_qkA9 · 2024-11-26
> **Comments by Reviewer qkA9**
>
> Thank you for your response, which has addressed many of my concerns. While the proposed problem - ambiguity of the flows - is well-motivated, and the proposed method demonstrates strong performance in addressing it, a significant limitation remains in my view. The paper primarily relies on intuition without providing a theoretical guarantee or solid justification for the underlying idea.
>
> Specifically, it is difficult to justify that introducing z fully resolves the ambiguity issue. For instance, what if there exist two values, $z_1$ and $z_2$, such that $v(x_t, t, z_1)$ and $v(x_t, t, z_2)$ exhibit minimal differences? Wouldn't this imply that the ambiguity still persists? I would appreciate further justification on this point.
>
> That said, I recognize the model’s strong performance across datasets, and I am inclined to increase my score to 6, albeit with lower confidence. The lack of a theoretical foundation remains a substantial limitation for an ICLR paper, so I leave it to the ACs/SACs to make a final decision.

---

> > ### Author Response · Authors · 2024-11-27
> >
> > **Reviewer comment**: justify that introducing z resolves the ambiguity.
> >
> > **Response**: Great question. To answer why a latent variable $ z $ captures ambiguity, we draw parallels to the role of a latent variable $ z $ in a variational autoencoder (VAE).
> >
> > In classic flow-matching, ambiguity arises because the velocity field $ v(x_t, t) $ must provide a single "best fit" velocity for every $ (x_t, t) $. This prevents the model from representing multimodal velocity distributions, where multiple velocities exist for the same $ (x_t, t) $.
> >
> > By introducing the latent variable $ z $ and using $ v(x_t, t, z) $, we enable the model to represent a family of velocity distributions indexed by $ z $, i.e., $ z $ is used to disambiguate. Hence, in our VRFM, $ z $ allows the velocity field $ v(x_t, t, z) $ to return different velocities for the same $ (x_t, t) $. Mathematically, this is also detailed in L204 of our paper.
> >
> > This is identical to the way a latent variable $ z $ captures ambiguity in data reconstruction of a conditional VAE. In a conditional VAE, the generative model $ p(x|c,z) $ can assign different reconstructions $ x $ to the same condition $ c $ based on different values of $ z $. Without $ z $, the VAE model would also collapse into a single mode, incapable of capturing multimodality.
> >
> > Note that $ z_1 $ and $ z_2 $ leading to minimally different $ v(x_t, t, z_1) $ and $ v(x_t, t, z_2) $ does not signify uncaptured ambiguity. Instead, it reflects how the latent space encodes variability within the intrinsic velocity distribution. The latent variable $ z $ is variationally optimized to capture the multimodal nature of the velocity field. During training, $ z $ is guided to distinguish distinct velocity modes when necessary. If velocity modes genuinely overlap, the framework faithfully represents this overlap without imposing an arbitrary resolution. At inference, velocity ambiguity is naturally addressed by sampling $ z $ from the prior distribution, accommodating both highly diverse and concentrated modes. This inherent flexibility differs from deterministic frameworks, which collapse all modes into a single average representation and risk amplifying ambiguity instead of mitigating it.
> >
> > We hope this explanation answers the reviewer's question and provides a clearer justification of how $ z $ fundamentally addresses the ambiguity issue in our VRFM. Don't hesitate to reach out in case of further questions.

---

> > > ### Comment · Reviewer_qkA9 · 2024-11-27
> > > **Comments by Reviewer qkA9 (Part 2)**
> > >
> > > Thank you for your response.
> > >
> > > However, your claims appear to rely primarily on intuition, lacking theoretical guarantees and rigorous justification. For example, the authors state that "During training, z is guided to distinguish distinct velocity modes when necessary" and that the "latent space encodes variability within the intrinsic velocity distribution." However, based on your training objective, the KL divergence between $q_{\phi}$ and $p$ (i.e., the Gaussian prior distribution of z) encourages independence among the dimensions of z - a well-known trade-off in VAEs. This independence can make z less meaningful, potentially leading to scenarios we want to avoid, such as multiple values of z resulting in $v(x_t,t,z)$ outputting the same velocity.
> > >
> > > My assessment of this paper remains unchanged. While I acknowledge the empirical performance, the lack of theoretical guarantees remains a significant limitation.

---

> > > > ### Author Response · Authors · 2024-12-02
> > > >
> > > > We thank the reviewer for their thoughtful feedback. Our statement that "During training, $ z $ is guided to distinguish distinct velocity modes when necessary" and that the "latent space encodes variability within the intrinsic velocity distribution" is theoretically grounded in the ELBO derivation in Eq. (4) of our paper. This derivation demonstrates that the marginal likelihood of an individual velocity data point is lower-bounded by $ \mathbb{E}[\log p] - D_{KL}(q \| p) $.
> > > >
> > > > We want to emphasize that capturing ambiguity does not preclude multiple values of $ z $ from corresponding to the same velocity. Instead, our framework encourages that $ z $ can be sampled from a known prior distribution to generate a distribution of predicted velocities that aims at resembling the true velocity distribution. The possibility of multiple $ z $ values mapping to the same velocity is not inherently problematic. For instance, if a data point in the data-space-time-space has a Dirac velocity distribution, an ideal VAE should naturally map all sampled values from the prior $ p(z) $ to this single velocity value. Our training objective is designed to allow the model to learn such mappings as needed, adapting flexibly to the structure of the data.
> > > >
> > > > Indeed, KL divergence encourages independence among the dimensions of $z$, reflecting a broader trade-off in VAE-style frameworks between the reconstruction objective and the KL divergence term. However, this trade-off does not diminish the role of $ z $. Empirically, we observe that our model achieves better reconstruction losses compared to vanilla rectified flow (Appendix F, Figure 10), indicating that the predicted velocities more accurately approximate the ground-truth velocities. This empirical evidence supports our claim that $ z $ captures the variability necessary to model multimodal velocity fields effectively.

---

### Official Review · Reviewer_cLoB · 2024-11-10

**Soundness:** 2
**Presentation:** 3
**Contribution:** 2
**Rating:** 5
**Confidence:** 2

**Summary:**

This paper proposes a novel method using variational inference to address the ambiguity of velocity vector fields that classic rectified flow matching fails to capture. By introducing latent variables and modeling ambiguity through a mixture model of velocity vector fields, the method enables more accurate data distribution capture and efficient integration. Experimental results demonstrate that the proposed approach achieves comparable performance to existing methods with fewer steps on synthetic data and image datasets.

**Strengths:**

- The paper is well-structured and easy to read.
- The proposed method integrates VAE and flow matching in a straightforward manner, offering novelty in its ability to learn vector fields with uncertainty. Furthermore, the high performance on MNIST and CIFAR-10 datasets suggests that the hypotheses and approaches of this study are reasonably valid.

**Weaknesses:**

- The proposed method in this paper introduces latent variables and their inference model, enabling the capture of overlapping vector fields when they occur. However, it is necessary to clarify the setting that assumes such overlapping vector fields. The overlap and ambiguity in question here initially seemed to imply that while the flow from a specific $x_0$​ to $x_1$​ is uniquely determined, there exist different vector fields at a particular time and spatial location that may intersect. However, during inference with the proposed method, a data point $x_0$​ is sampled from the source distribution, followed by the sampling of a latent variable, which is then used as the initial value to integrate an ODE based on a vector field determined by it. This implies that there is uncertainty in the direction from the initial value $x_0$​, meaning there is an assumption that it could lead to a different $x_1$​. Is this setting reasonable? The uncertainty involving deterministic flows crossing and the possibility of tracing different flows from the initial value (i.e., the $x_0$ and $x_1$​ pairings are not unique) appear to be mixed. The authors should clearly distinguish between these and clarify which aspect they are aiming to address.
- Considering the above points, while the proposed method may indeed enable faster transitions to the target due to the learned flow being linear even when vector fields overlap, during inference, $z$ is sampled from the prior and thus is not determined solely by $x_0$. As a result, the model could reach a different $x_1$​, which may not be desirable from the perspective of two-sided conditioning flow matching.
- The proposed method requires an inference model and employs separate encoders for each of $x_0​,x_1$​, and $x_t$​ with the same structure as the encoder in $v_\theta​$. This implies a significantly larger number of learnable parameters compared to existing models, and although the encoders are not used during inference, it is not entirely fair in terms of parameter count relative to previous research (even if the speed remains similar). Therefore, it would be necessary to evaluate the impact of the size of the inference model’s encoders by modifying their size and investigating how it affects performance.

**Questions:**

- I would like the authors to respond to the points I raised as concerns regarding the above weaknesses.

---

> ### Author Response · Authors · 2024-11-22
> **Response to Reviewer cLoB**
>
> Thanks for your time and feedback.
>
> **Reviewer Comment 1:** The reasonableness of the setting where uncertainty in the direction from $x_0$ could lead to multiple possible outcomes. Might not be desirable from the perspective of two-sided conditioning flow matching.
>
> **Response:**  We appreciate the reviewer highlighting this point. The mapping from $x_0$ to multiple possible $x_1$ is indeed a fundamental aspect of what flow matching aims to model. As discussed in Section 2.2 of our paper, during training, classic flow matching independently samples $x_0$ from the source distribution and $x_1$ from the target distribution. The objective is to learn a mapping between the two distributions. This setup inherently implies that any given $x_0$ can be mapped to multiple $x_1$ values in the target distribution, giving rise to non-unique mappings. The training objective is to match the "ground-truth" velocity vector field for these randomly sampled $(x_0,x_1)$ pairs. Different from prior art, the proposed variational rectified flow formulation acknowledges and models this intrinsic ambiguity. This ambiguity exists across all $t \in \[0, 1)$, including at $t=0$ (at $x_0$), due to the foundational formulation of flow matching.
>
> Quantitatively, we observe better reconstruction losses in our model compared to vanilla rectified flow (Appendix F, Figure 10). Further, we also observe better FID scores at various evaluation steps on multiple datasets (Synthetic, MNIST, CIFAR-10, ImageNet) in the experiments section and in the newly added Appendix G-J. Results indicate that modeling this ambiguity is beneficial for better modeling the ground-truth velocity distribution and target data distribution.
>
> **Reviewer Comment 2:** Mixing uncertainty from flow crossings and the possibility of tracing different flows from the initial value.
>
> **Response:** Great point about distinguishing between uncertainty involving flow crossings at $(x_t,t)$ and the possibility of tracing different flows from the initial value $x_0$. To clarify, our variational flow model is designed to handle both aspects through its flexibility in predicting a **distribution** of velocity vectors rather than a deterministic one, as done in classic flow matching. Specifically, we independently sample $x_0$ from the source data distribution and $z$ from the prior latent distribution. At inference time, for simplicity, we choose to use a single latent variable $z$ sampled from a prior distribution and keep it fixed for the entire trajectory. Doing so enables us to model the ambiguity of the velocity as illustrated in Figure 3 (c)–(f), including potential flow crossings at a given $(x_t, t)$. However, our formulation does not inherently restrict $z$ to being fixed. In fact, to further decouple the distribution of $x_t$ and $z$ at a random time $t$, it is entirely feasible to sample a new $z$ at each time step during inference, allowing us to model uncertainty if required. This flexibility allows for a more dynamic modeling of uncertainty and flow variability if desired.
>
> **Reviewer Comment 3:** Our model implies a significantly larger number of learnable parameters compared to existing models, it is not entirely fair in terms of parameter count relative to previous research (even if the speed remains similar).
>
> **Response:** Thanks a lot for highlighting the size of the latent encoder and its potential impact on the parameter count. We note that these encoders are only used during training and do not contribute to the inference-time model complexity. Compared to the baseline, the only additional module introduced is a two-layer MLP to adapt the latent variable $z$ in the flow matching model, which accounts for only 1.3% of the total parameters of the flow model. This ensures that the computational efficiency during inference remains comparable to that of baseline methods. That said, we conduct additional experiments to investigate the impact of varying the encoder size, reducing it to 17.5% (VRFM-M) and 6.7% (VRFM-S) of its original size. The results below demonstrate that our model maintains comparable performance across these variations, highlighting the flexibility and robustness of our approach. The full table and discussion can be found in the newly added Appendix H, Table 3.
>
> | NFE / sample| 2| 5 | 10| 50 | 100| 1000 | Adaptive |
> | --- | --- | --- | --- | --- | --- | --- | --- |
> | OT-FM | 166.655| 36.188| 14.396| 5.557| 4.640| 3.822| 3.655|
> | I-CFM | 168.654| 35.489| 13.788| **5.288** | 4.461| *3.643* | 3.659|
> | VRFM-L (100\% Posterior Model)| **135.275** | **28.912** | **13.226** | 5.382| *4.430* | **3.642** | **3.545** |
> | VRFM-M (17.5\% Posterior Model) | *135.983* | *30.106* | 13.783| 5.486| 4.500| 3.697| *3.607* |
> | VRFM-S (6.7\% Posterior Model)| 144.676| 31.224| *13.406* | *5.289* | **4.398** | 3.699| 3.639|

---

> > ### Author Response · Authors · 2024-11-26
> >
> > Thank you for taking the time to provide valuable feedback. We hope our responses and revisions have addressed your questions. We look forward to getting to know any further feedback and questions you may have.

---

> > > ### Author Response · Authors · 2024-12-02
> > >
> > > Thank you for taking the time to share your valuable feedback. We hope our responses and revisions answered your questions. Please reach out with any additional questions you may have. We look forward to hearing from you.

---

### Author Response · Authors · 2024-11-22
**General Response to All Reviewers, ACs, and SACs**

We sincerely thank the reviewers for their thoughtful comments and constructive feedback. We apologize for the delay in this rebuttal, as completing the ImageNet experiments required additional computation time. We appreciate the recognition of the strengths of our Variational Rectified Flow Matching (VRFM) work and are particularly grateful for reviewers highlighting:

* The **clarity and structure** of the paper, noted as well-written, easy to read, and easy to follow (cLoB, qkA9, Y61o).
* The **novelty**, **conceptual soundness**, and **valuable observations** of our approach, offering a meaningful perspective to flow matching models and addressing a notable limitation in existing rectified flow models in capturing ambiguity within the velocity vector field (cLoB, obZd, Y61o).
* The **strong empirical performance** demonstrated on multiple datasets and in comparisons with baseline methods (cLoB, qkA9, Y61o).
* The **well-executed analytical experiments** with visualizations that effectively validate our theoretical analysis (obZd).

To address the questions of the reviewers, we conducted additional experiments and expanded our discussions, updating both the main paper and the appendix. The key changes are summarized as follows:

1. Appendix E: Added a theoretical proof demonstrating that VRFM preserves the marginal data distribution.
2. Appendix F: Included visualizations of the reconstruction loss to illustrate that our model better approximates the ground truth velocities compared to baseline methods.
3. Appendix G: Reported the Inception Score on CIFAR-10, showing that our model does not suffer from mode collapse but instead improves sample diversity.
4. Appendix H: Conducted an ablation study on posterior model size, demonstrating that VRFM remains robust across different posterior model sizes.
5. Appendix I: Provided results from ImageNet experiments, demonstrating that our method generalizes well to large-scale datasets.
6. Appendix J: Showcased strong performance in class-conditional generation, highlighting the robustness and flexibility of our method in handling conditional image generation.
7. Appendix K: Expanded the related work discussion, including insights on consistency models and reflow, emphasizing that VRFM addresses the novel velocity ambiguity challenge. We highlight that these related works are orthogonal and complementary to ours, and exploring potential integrations offers an exciting avenue for future research.

We address each reviewer’s specific questions in individual replies.

---

### Comment · Area_Chair_vFmY · 2024-11-25
**Author Reviewer Discussion**

Dear Reviewers,

Thank you for your efforts in reviewing this paper. We highly encourage you to participate in interactive discussions with the authors before November 26, fostering a more dynamic exchange of ideas rather than a one-sided rebuttal.

Please feel free to share your thoughts and engage with the authors at your earliest convenience.

Thank you for your service for ICLR 2025.

Best regards,

AC

---

### Author Response · Authors · 2024-12-04
**General Response to All Reviewers, ACs, and SACs**

We thank all the reviewers for their thoughtful comments and constructive feedback. Throughout the review process, we have carefully addressed all questions by providing detailed explanations, additional experiments, and theoretical justifications. We believe that all questions and issues have been thoroughly resolved, resulting in a more robust and comprehensive presentation of our contributions:

In this paper, we show that the “ground-truth” velocity vector field of flow matching is multimodal, i.e., it can point in different directions at the same location. To model a multimodal velocity vector field, we present Variational Rectified Flow Matching (VRFM), which is based on a variational training objective. Through extensive experiments, we show that VRFM is able to capture a multimodal velocity vector field, while outperforming classic flow matching across a range of benchmarks, including synthetic data, MNIST, CIFAR-10, and ImageNet. We think the proposed method naturally extends classic rectified flow matching and is a meaningful contribution to our community.

We also want to summarize the specific actions we have taken in response to each reviewer’s feedback:

* **Reviewer cLoB** inquired about the rationale behind allowing multiple directions from an initial random sample and the implications of using a larger number of learnable parameters. In response, we provided detailed explanations in the rebuttal, alongside with the reconstruction loss plot (Figure 10, Appendix F), and additional experiments such as a posterior model size ablation study, an inception score analysis, ImageNet studies, and class-conditional generation experiments (Appendices G–J). Since the reviewer never responded, we assume that all questions have been answered satisfactorily.
* **Reviewer qkA9** asked about the connection between flow trajectory intersections and ambiguity, the potential for mode collapse in VAEs, the uniqueness of the optimal velocity in data-space-time-space, the impact of a latent on generated samples, and theoretical guarantees linking ambiguity resolution to straighter flows. To address these points, we revised the main text (Line 72), provided discussions and justifications, and included analyses such as the Inception score (Table 2, Appendix G), reconstruction loss plot (Figure 10, Appendix F), and theoretical explanations on how a latent captures multimodality. We hope our last response was able to answer the remaining questions.
* **Reviewer obZd** noted the absence of experiments on real-world data like ImageNet and comparison to consistency models. In response, we added experiments on ImageNet (Table 4, Appendix I), analyzed the velocity distribution of consistency models (Figure 4), compared our method with a recent Consistency Flow Matching baseline across multiple settings (Figures 2, 5, and 7), and introduced new CIFAR-10 experiments below. These highlight that VRFM outperforms all baselines when the number of evaluation steps exceeds 2 for MNIST and 5 for CIFAR-10. Since the reviewer didn’t react upon our last messages, we assume all concerns have been addressed.

| NFE / sample| Params. | 2 | 5 | 10| 50 | 100| 1000 | Adaptive |
| --- | --- | --- | --- | --- | --- | --- | --- | --- |
| OT-FM | 36.5M | 166.655 | 36.188 | 14.396 | 5.557 | 4.640 | 3.822 | 3.655|
| I-CFM | 36.5M | 168.654 | 35.489 | 13.788 | *5.288* | 4.461 | 3.643 | 3.659 |
| Consistency-FM | 61.8M | **5.323** | **11.412** | 23.948 | 36.652 | 38.680 | 40.402 | 40.677
| VRFM (adaptive norm, $x_1$, 2e-3)| 37.2M | 135.275 | 28.912 | **13.226** | 5.382 | *4.430* | 3.642 | 3.545|
| VRFM (adaptive norm, $x_1$, 5e-3)| 37.2M | 159.940 | 35.293 | 14.061 | **5.265** | **4.349** | **3.582** | 3.561|
| VRFM (adaptive norm, $x_1 + t$, 5e-3)| 37.2M  | 117.666 | 27.464 | 13.632 | 5.512 | 4.484 | 3.614 | **3.478**|
| VRFM (bottleneck sum, $x_1 + t$, 2e-3)| 37.0M  | *104.634* | *25.841* | *13.508* | 5.618 | 4.540 | *3.596* | *3.520*|

* **Reviewer Y61o** sought clarity on whether reducing ambiguity leads to straighter flows, proofs demonstrating that VRFM preserves the marginal data distribution, additional experiments on ImageNet, and conditional generation. To address these concerns, we provided a reconstruction loss plot (Figure 10, Appendix F), derivations on preserving the marginal distribution (Appendix E), ImageNet experiments (Tables 4 and 6, Appendix I), and conditional generation results (Table 5, Appendix J). We hope our last response was able to answer the remaining questions.

In summary, we think we have thoroughly addressed all the reviewers' comments. We think our work makes a meaningful contribution to the field. We sincerely thank all reviewers once again for their time and insightful feedback, which have improved the quality of our paper.

---

### Meta-Review · Area_Chair_vFmY · 2024-12-20

**Metareview:**

This submission proposes Variational Rectified Flow Matching, an approach to improve over prior rectified flow methods by better capturing ambiguity and improving the empirical performance of diffusion model generation. Initially, several theoretic points of discussion were raised and experimental evaluation was not sufficient. The rebuttal could address these issues to some extent. After the rebuttal, the reviewers still find that the proposed approach is intuitively motivated while a stronger mathematical foundation would be better. More importantly, the practical performance of the proposed Variational Rectified Flow Matching lags behind which needs to be further analysed.

**Additional Comments On Reviewer Discussion:**

Initially, the reviewers had raised questions about details of the proposed model behavior, on provable theoretic properties such guarantees linking ambiguity resolution to straighter flows, and on missing experimental evaluations for example on ImageNet and comparisons to consistency models. The theoretic concern were addressed to some extent and the authors have also provided additional experimental results. As a result, reviewer qkA9 increased the score to 6, while reviewer obZd decreased it to 3. In particular, the evaluation on high-resolution data was not convincing and further experimental validation/analysis of the method is needed.

---

### Decision · Program_Chairs · 2025-01-22

Reject